# FedLLM-Bench: Realistic Benchmarks for Federated Learning of Large Language Models

**Rui Ye[1]*** **Rui Ge[1]*** **Xinyu Zhu[1]** **Jingyi Chai[1]** **Yaxin Du[1]**
**Yang Liu[2]** **Yanfeng Wang[3,1]** **Siheng Chen[1]†**

[1] Shanghai Jiao Tong University
[2] Institute for AI Industry Research, Tsinghua University    [3] Shanghai AI Laboratory

## Abstract

Federated learning could enable multiple parties to collaboratively fine-tune large language models without directly sharing their data (FedLLM). Following this training paradigm, the community has put massive efforts from diverse aspects including framework, performance, and privacy. However, an unpleasant fact is that there are currently no realistic datasets and benchmarks for FedLLM and previous works often rely on artificially constructed datasets, failing to capture properties in real-world scenarios. Addressing this, we propose FedLLM-Bench, which involves 8 training methods, 4 training datasets, and 6 evaluation metrics, to offer a comprehensive testbed for the FedLLM community. FedLLM-Bench encompasses three datasets (e.g., user-annotated multilingual dataset) for federated instruction tuning and one dataset (e.g., user-annotated preference dataset) for federated preference alignment, whose scale of client number ranges from 38 to 747. Our datasets incorporate several representative diversities: language, quality, quantity, instruction, length, embedding, and preference, capturing properties in real-world scenarios. Based on FedLLM-Bench, we conduct experiments on all datasets to benchmark existing FL methods and provide empirical insights (e.g., multilingual collaboration). We believe that our FedLLM-Bench can benefit the FedLLM community by reducing required efforts, providing a practical testbed, and promoting fair comparisons. Code and datasets are available at https://github.com/rui-ye/FedLLM-Bench.

## 1 Introduction

Large language models (LLMs) have achieved unprecedented success in diverse domains [1, 2, 3, 4, 5, 6]. These LLMs are usually trained by centralized learning paradigm, where various parties individually collect massive data for model training. In this case, the data amount of each individual party is hard to scale due to the high cost of collecting and annotating data. However, their data cannot be directly shared for collaboration due to property and privacy issues.

To relieve the required cost of each party, federated learning (FL) [7, 8] has emerged as a sound and off-the-shelf technique to facilitate collaboration, which leverages decentralized language data to collaboratively train LLMs in a privacy-preserving way (FedLLM)[1] [9, 10, 11]. To facilitate the development of FedLLM, there has been a series of code frameworks such as OpenFedLLM [9], FederatedScope-LLM [10], and FedML-LLM [11]; and many methods that tackle the issues of data quality [12], safety risks [13], incentive mechanism [14], intellectual property protection [15], privacy [16], limited resources [17], data and system heterogeneity [18] in FedLLM.

---

* Equal contributions. † Corresponding author: sihengc@sjtu.edu.cn.
[1]We are referring to post-training of LLMs, which is more suitable in FL considering the required resources.

Table 1: Summary of our four realistic FedLLM datasets. IT denotes instruction tuning and PA denotes preference alignment. # denotes 'the number of' and L. denotes 'the length of'. Our datasets exhibit diversities in characteristic, task, client number, quantity, length, and quality. See statistics of unfiltered versions in Table 10.

| Dataset Name | Fed-Aya [23] | Fed-ChatbotIT [24] | Fed-WildChat [25] | Fed-ChatbotPA [24] |
|---|---|---|---|---|
| Characteristic | Multilingual | Single-Turn chat | Multi-Turn chat | Preference |
| Applied Task | IT | IT | IT | PA |
| # Clients (Total) | 38 | 237 | 100 | 747 |
| # Samples (Total) | 25,513 | 6,166 | 52,703 | 9,508 |
| # Samples (Client) | $671 \pm 815$ | $26 \pm 33$ | $527 \pm 477$ | $13 \pm 21$ |
| L. Instruction (Client) | $116 \pm 199$ | $68 \pm 119$ | $331 \pm 435$ | $69 \pm 124$ |
| L. Response (Client) | $225 \pm 411$ | $211 \pm 176$ | $506 \pm 470$ | $218 \pm 178$ |
| Data Quality (Client) | $0.63 \pm 0.28$ | $0.67 \pm 0.22$ | $0.79 \pm 0.37$ | $0.68 \pm 0.21$ |

Despite that massive efforts have been made, one significant concern remains: there is currently no realistic benchmark for FedLLM, making it hard to practically evaluate the effectiveness of FL methods in real-world scenarios. In such context, each previous work constructs its own FL datasets by artificially partitioning existing centralized datasets [9, 10, 15], falling short of capturing the natural properties existed in real-world cross-user datasets [19, 20]. Even worse, these papers often follow different training and evaluation setups, which significantly increases the difficulty of re-implementations and risk of unfair comparisons [21, 16].

To fill this gap, we propose the first realistic benchmark for FedLLM termed FedLLM-Bench, offering a comprehensive testbed for the FedLLM community. FedLLM-Bench encompasses three datasets for federated instruction tuning (including one user-annotated multilingual dataset: Fed-Aya, and two datasets with realistic user instructions: Fed-WildChat and Fed-ChatbotIT) and one dataset (user-annotated preference dataset: Fed-ChatbotPA) for federated preference alignment. These datasets are all naturally split by real-world user ID with the scale ranging from 38 to 747 clients, therefore exhibiting realistic federated properties (especially for cross-device setup in FL where data are partitioned by user devices) [20, 19]. Specifically, datasets in our FedLLM-Bench inherit the following diversities (Table 1): **(1) Language:** clients' datasets (e.g., our Fed-Aya dataset) cover data from diverse languages, modeling the real-world scenarios of multilingual collaboration (see Figure 1(a)). **(2) Quality and Quantity:** the quality and quantity of clients' datasets vary across each other, which is a common property in real-world applications; see detailed illustrations in Figure 3 and 5. **(3) Length:** the sequence length of clients' data could be quite different, representing a new type of data heterogeneity in FL; see Figure 1(b). **(4) Preference:** different clients have different preferences as verified by different preferred instructions in instruction tuning datasets (e.g., Fed-WildChat) and different preferred responses in preference alignment dataset (i.e., Fed-ChatbotPA), mirroring the complexities of real-world data scenarios; see detailed illustrations in Figure 1(c), 2, 1(d). These diversities make our FedLLM-Bench a comprehensive benchmark in the era of FedLLM, serving as a great successor to representative benchmarks for classical tasks such as LEAF [22] benchmark.

Based on these datasets, we implement 8 representative baseline methods and 6 evaluation metrics, and conduct extensive experiments. Our experiments mainly demonstrate (1) that federated learning can consistently bring performance gain compared to local training without collaboration; and (2) the performance ranking of several representative baseline methods. Besides serving as a benchmark for performance comparison, our FedLLM-Bench can also support exploration of new research directions thanks to its flexibility and diversity. As an example, we conduct an exploratory experiment based on the multilingual dataset Fed-Aya, showing that collaboration among similar languages could potentially bring more benefits comparing to collaboration among all languages.

Our contributions are as follows:

1. We propose the first realistic benchmark for federated post-training of LLMs, FedLLM-Bench, which encompasses four naturally split datasets. FedLLM-Bench covers diverse tasks, scales, languages, qualities, quantities, lengths, and preferences, mirroring the complexities and diversities of real-world scenarios.

2. We integrate these datasets into a codebase with 8 representative baseline methods and 6 evaluation metrics, and open-source the datasets with the integrated codebase for the community.

3. We conduct extensive experiments to demonstrate the status of several existing baseline methods on our FedLLM-Bench and show its potential in promoting exploration of new research directions.

## 2 Related work

**Federated learning for large language models.** Federated learning is a privacy-preserving and collaborative training paradigm that enables multiple parties to collaboratively train a shared global model without sharing their raw data [7, 8]. Data heterogeneity is one of the most representative challenges in FL, where clients' datasets are drawn from different distributions. Addressing this, numerous methods have been proposed by regularization [26], gradient correction [27], feature alignment [28], adjustment of aggregation weights [29, 30], or introducing momentum [31, 32].

Recently, having witnessed the success of large language models (LLMs) in centralized learning [1, 33, 34, 3, 35], many researchers start to explore training LLMs via federated learning, mitigating the issue of the shortage of public data or private data of one individual [2, 36, 37, 9]. Within one year, there have been many frameworks such as OpenFedLLM [9], FederatedScope-LLM [10], FedML-LLM [11], and diverse works such as attacks of safety alignment in FedLLM [13], FedbiOT [15] that protects model property, FFA-LoRA [16] that improves performance under differential privacy, HetLoRA [18] that addresses data and system heterogeneity problem, iPFL that focuses on incentive mechanism.

However, one significant issue of these previous works is that their experiments are all based on artificially crafted FL datasets, falling short of extrapolating their effectiveness in real-world scenarios. Addressing this, we propose the first realistic benchmark for FedLLM, FedLLM-Bench, which mirrors the complexities and diversities in real-world applications. Besides, we implement 8 representative baseline methods in our FedLLM-Bench to demonstrate their effectiveness in realistic scenarios.

**Datasets and benchmarks in federated learning.** Since clients' data are collected independently without coordination, the issue of data heterogeneity commonly exists in FL. A large proportion of FL works [7, 29, 26, 28] simulate data heterogeneity by artificially partitioning classical datasets such as CIFAR-10/100 [38], Fashion-MNIST [39], and MNIST [40]. Recently, researchers on FL and pre-training of LLMs may split dataset according to tags such as web domain, article, or book [41]; while researchers on FL and post-training of LLMs may artificially partition centralized dataset based on task type [14, 42] or coding language [10]. Addressing this, several realistic benchmarks are proposed for classic tasks such as image and text classification, which include LEAF [22] (a suite of user-split datasets), FLAIR [20] (multi-label image classification), and FLamby [19] (a benchmark for medical analysis). However, currently, there is no realistic dataset or benchmark for the tasks of post-training of FedLLM, while our FedLLM-Bench stands out as the first one in the literature. Besides, our FedLLM-Bench covers two unique post-training tasks compared to previous benchmarks: federated instruction tuning [43, 44, 9] and federated preference alignment [45, 46, 9].

## 3 FedLLM-Bench: a realistic benchmark for FedLLM

Here, we introduce our FedLLM-Bench, from four perspectives: training methods, descriptions of training datasets, analysis of training datasets, and evaluation metrics.

### 3.1 Training methods

**FedLLM overview.** FedLLM involves four iterative steps: server-to-client model downloading, local model training, client-to-server model uploading, and global model aggregation. During FedLLM, clients could collaborate on two critical tasks for LLMs [2]: instruction tuning and preference alignment, which are challenging for individuals due to high cost of data collection [47, 48]. Besides, various FL baseline methods [26, 27, 32] can be incorporated into FedLLM.

**Tasks: instruction tuning & preference alignment.** In instruction tuning [2, 9], each data sample is an instruction-response pair, where the LLMs are trained to follow instructions to generate the expected responses via supervised fine-tuning. In preference alignment [49, 9], each data sample consists of an instruction, a preferred and a dispreferred response, where the LLMs are trained to align with the preferred response given user instructions via direct preference optimization [46].

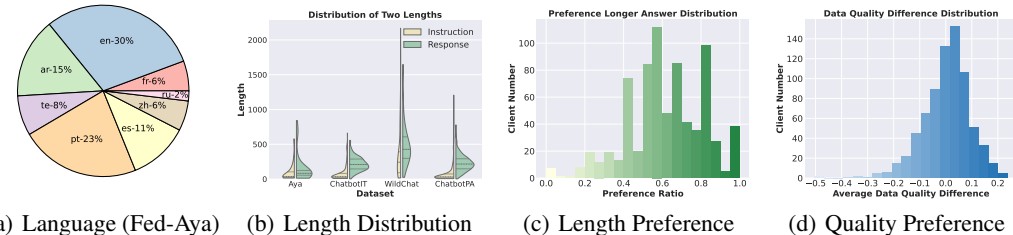

(a) Language (Fed-Aya)    (b) Length Distribution    (c) Length Preference    (d) Quality Preference

Figure 1: (a) Langauge distribution of clients in Fed-Aya dataset. (b) The distribution of length of instruction and response of clients' data. (c) Distribution of length preference (the ratio of a user preferring longer response) of clients in Fed-ChatbotPA dataset. (d) Distribution of quality preference (quality difference between preferred and dispreferred data) of clients in Fed-ChatbotPA dataset.

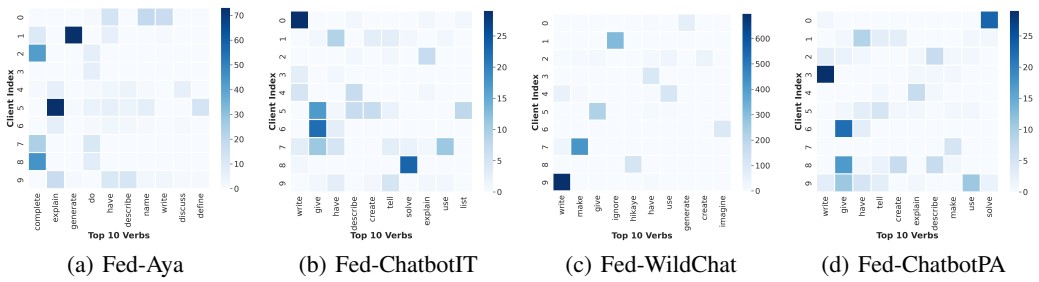

(a) Fed-Aya      (b) Fed-ChatbotIT      (c) Fed-WildChat      (d) Fed-ChatbotPA

Figure 2: Distributions of top 10 verbs in instructions (10 clients are plotted for illustration). Our realistic FedLLM datasets exhibit diverse patterns with respect to instruction types.

For both two tasks, we adopt the most commonly used parameter-efficient fine-tuning technique LoRA [50], reducing the requirements of computation and communication in FL [51, 52].

**FL: baseline methods.** In our FedLLM-Bench, we implement 8 representative baseline methods, including local training without collaboration and 7 classical FL baseline methods. Following the standard baseline FedAvg [7], at the local training part, we implement FedProx [26] which applies local-global model regularization and SCAFFOLD [27] which introduces control variate to correct local gradients; while at the model aggregation part, we implement FedAvgM [31], FedAdagrad [32], FedYogi [32], and FedAdam [32] which introduce model momentum to update global model.

### 3.2 Descriptions of training datasets

**Fed-Aya.** Aya [23] dataset is a multilingual instruction tuning dataset annotated by contributors from various countries [53]. We select 6 high-resource languages: English (en), Spanish (es), French (fr), Russian (ru), Portuguese (pt), Chinese (zh), and 2 low-resource languages: Standard Arabic (ar) and Telugu (te). According to the annotator ID, we filter out those who contribute less than 100 annotations, and construct Fed-Aya, which consists of 38 clients with 25k data samples in total. This dataset models a real-world federated scenario [54] where collaborating clients are distributed around the globe and aim to advance multilingual LLMs [1, 55]. We visualize the language distribution of Fed-Aya dataset in Figure 1(a), showing that the number of clients for different languages varies. Therefore, it also provides a dataset basis for the explorations of new research topics in FedLLM, including language personalization and fairness across high- and low-resource languages.

**Fed-ChatbotIT.** Chatbot-Arena-Conversations [56] is originally a collection of human-annotated preference data, where each data sample consists of a user instruction, a user-chosen response and a user-rejected response. Here, for each data sample, we combine the instruction and user-chosen response as an instruction-response pair. Subsequently, according to the user ID of the annotator, we filter out those who contribute less than 10 data samples and construct Fed-ChatbotIT, which consists of 237 clients with 6k data samples in total. This dataset captures the diversities of realistic use cases in single-turn query of LLMs, where instructions of different users could hold different patterns.

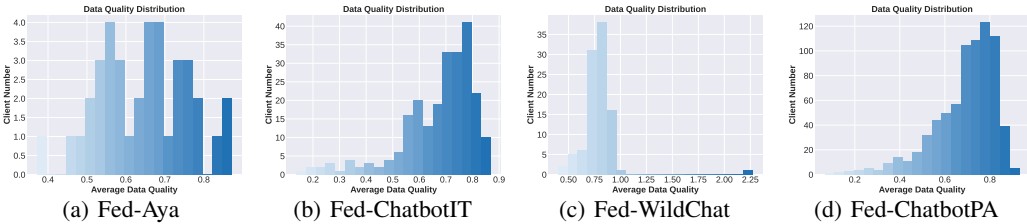

|     |     |     |     |
| :-: | :-: | :-: | :-: |
| (a) Fed-Aya | (b) Fed-ChatbotIT | (c) Fed-WildChat | (d) Fed-ChatbotPA |

Figure 3: The dataset quality distribution of clients in four training datasets: Fed-Aya, Fed-WildChat, Fed-ChatbotIT and Fed-ChatbotPA. We average the IFD scores of all instruction-response pairs of each client to represent the client's dataset quality.

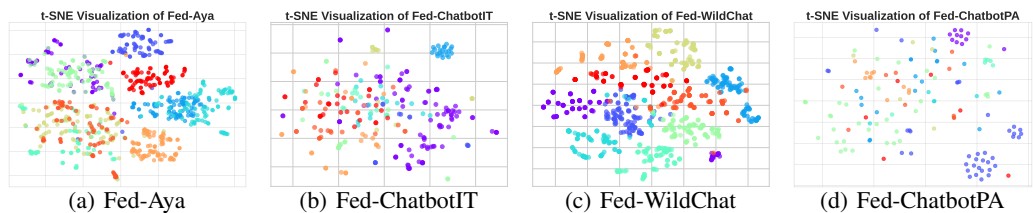

|     |     |     |     |
| :-: | :-: | :-: | :-: |
| (a) Fed-Aya | (b) Fed-ChatbotIT | (c) Fed-WildChat | (d) Fed-ChatbotPA |

Figure 4: The t-SNE visualization of embeddings of instruction-response pairs from 10 clients in Fed-Aya, Fed-ChatbotIT, Fed-WildChat, and Fed-ChatbotPA datasets. Each color denotes one client. We can see clustering phenomenon of one client's data and that clients' data are diverse.

**Fed-WildChat.** WildChat [25] is a collection of conversations between humans and ChatGPT, which contains a broad spectrum of user-chatbot interactions. According to the IP address, we partition the whole dataset into several user datasets and filter out those with less than 200 samples, forming our Fed-WildChat. Fed-WildChat consists of 100 clients with 53k data samples in total. This dataset represents real-world use cases between humans and chatbots, which involve multi-turn interactions.

**Fed-ChatbotPA.** We construct another federated version of Chatbot-Arena-Conversations [24] for preference alignment tasks: Fed-ChatbotPA. Specifically, we filter out users who contribute fewer than 5 preference pairs and the resulting Fed-ChatbotPA consists of 747 clients with 10k data samples in total. Each data sample contains a user instruction, a preferred and dispreferred response. This dataset exhibits real-world property that different individuals could have different preferences. To verify this, we analyze the dataset from two perspectives. Firstly, we visualize the length preferences of clients in Figure 1(c), where for each client we compute the ratio of the preferred responses being longer than the dispreferred responses. We see that most clients tend to prefer longer responses (i.e., the ratio is larger than 0.5) and clients have various preference ratios. Secondly, we visualize clients' quality preferences in Figure 1(d), where for each client we compute the averaged quality difference between the preferred and dispreferred data. We can see the diversity of clients' quality preferences.

Drawing inspiration from centralized learning of LLMs that the diversity of training data is critical for post-training of LLMs, we filter out clients with few data samples in the above construction process. However, to facilitate more comprehensive research, we also release the unfiltered versions, see statistics in Table 10.

### 3.3 Analysis of training datasets

We further show the diversities of our datasets for FedLLM from four perspectives.

**(1) Length.** For each client, we tokenize the instruction and response of each data sample using tokenizer of Llama2 [34], and average their length respectively. We plot the length distribution of clients in Figure 1(b). We can see that clients' data varies in data length and different datasets exhibit different distributions, verifying both inter-dataset and intra-dataset diversities. **(2) Instruction.** Following Self-Instruct [57], we use the Berkeley Neural Parser [58] to parse the instructions and extract the root verbs. We randomly sample 10 clients for each dataset and visualize the distribution of top-10 verbs in Figure 2. We can see that clients have different usage preferences in their instructions

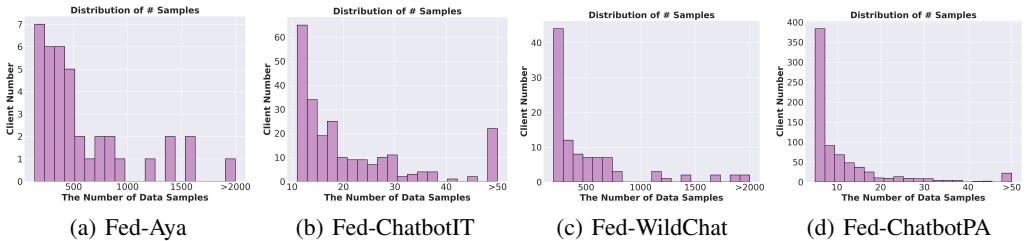

|  (a) Fed-Aya | (b) Fed-ChatbotIT | (c) Fed-WildChat | (d) Fed-ChatbotPA |

Figure 5: Data quantity distribution across clients of our four FedLLM datasets. We can see a variety of data quantities of clients, where a large proportion of clients have relatively few data.

and that the top 10 verbs vary among datasets. See more detailed visualizations in Figure 7, 9, 10, 11, 12. **(3) Quality.** We measure the data quality using IFD metric [59], where a higher value denotes higher instruction-following difficulty, and average the quality of all data samples of one client. From Figure 3, we can observe the diversities in clients' data, and distinct distributions among these four datasets. **(4) Embedding.** We randomly sample 10 clients, extract the embedding of each instruction-response data sample using text-embedding-ada-002 model from OpenAI, and plot them via t-SNE [60] in Figure 4, where each color denotes one client. We can see that there is clustering phenomenon, indicating certain patterns within one client's data that mirror real-world cases. It also demonstrates the diversity of clients' data. **(5) Quantity.** We plot clients' data quantity distribution in Figure 5. From the figure, we see a variety of data quantity across clients for all datasets, where most clients have relatively fewer data samples.

## 3.4 Evaluation metrics

To evaluate the effectiveness of the training methods on our realistic FL datasets, we consider 6 evaluation metrics, including 4 open-ended metrics and 2 close-ended metrics.

**Open-ended evaluation.** MT-Bench [56] is one of the most acknowledged evaluation metrics in the era of LLMs, which evaluates both one-turn and two-turn conversation capability. Similarly, Vicuna bench [61] evaluates one-turn instruction-following capability. AdvBench [62] evaluates the safety rate given unsafe instructions. Additionally, we consider an in-domain evaluation metric termed Ref-GPT4, where we randomly sample 50 unseen data as the test set and use GPT-4 to rate the generated response given the ground-truth response as reference [63, 56]; see prompt in Figure 8.

**Close-ended evaluation.** We consider two common close-ended evaluations [64, 34, 35, 1, 9]: MMLU [65] (measuring knowledge of 57 subjects) and HumanEval [66] (measuring capability of generating programs from docstrings). We evaluate these two metrics mainly to ensure that fine-tuning will not compromise these capabilities acquired during pre-training.

## 4 Experiments on FedLLM-Bench

**Experimental setups.** For instruction tuning task, we use Llama2-7B [34] as the base model and set the learning rate as $2e^{-5}$ with a batch size of 16. For preference alignment task, we use Alpaca-7B [67] as the base model and set the learning rate $1e^{-4}$ with a batch size of 8, as a large proportion of clients have fewer than 10 data samples. We adopt 8-bit quantization on the base model and set the rank of LoRA [50] as 16. The number of communication rounds is set to either 100 or 200 and only a small proportion of clients are sampled for each round. We set the number of steps of local model training as 10 for most scenarios. The chosen hyper-parameters of baseline methods are shown in Table 12. Please refer to more details in Appendix C.1.

### 4.1 Benchmark results

**Fed-Aya.** Here, we conduct experiments on Fed-Aya with Ref-GPT4 as the evaluation metric for 8 languages. We run local training for each language (randomly sample one client for simplicity), and 7 federated methods. From Table 2, we see that: (1) Most FL methods can outperform local training on average, indicating the effectiveness of collaboration. (2) No FL method can achieve

Table 2: Experiments on multilingual dataset Fed-Aya evaluated via Ref-GPT4. FL methods generally perform better than local training on average. However, FL methods can not ensure better performance on every language, implying the necessity for exploring language personalization techniques.

| Algorithm | ar | en | es | fr | pt | ru | te | zh | Average |
|---|---|---|---|---|---|---|---|---|---|
| Local Training (ar) | 2.55 | 7.55 | 4.85 | 5.10 | 3.95 | 4.55 | 1.55 | 3.35 | 4.18 |
| Local Training (en) | 2.55 | 7.20 | 5.35 | 4.60 | **5.35** | 4.75 | 1.60 | 3.55 | 4.37 |
| Local Training (es) | 1.90 | 7.80 | 5.55 | 5.60 | 4.50 | 5.20 | 1.30 | 5.05 | 4.62 |
| Local Training (fr) | 1.85 | 7.90 | 4.75 | 4.20 | 4.25 | 5.05 | 1.30 | 3.95 | 4.16 |
| Local Training (pt) | 1.95 | 5.95 | 4.20 | 5.45 | 3.85 | 5.15 | 1.55 | 3.95 | 4.01 |
| Local Training (ru) | 1.60 | 7.80 | 6.05 | 4.80 | 4.00 | 4.50 | 1.75 | 4.90 | 4.43 |
| Local Training (te) | 2.10 | 3.70 | 3.75 | 3.50 | 3.05 | 4.10 | 1.25 | 3.60 | 3.13 |
| Local Training (zh) | 2.30 | 8.10 | 5.45 | 5.80 | 4.80 | 4.30 | 1.60 | 4.95 | 4.66 |
| FedAvg [7] | 2.50 | 8.00 | 5.50 | 5.35 | 4.95 | 5.65 | **2.00** | 5.25 | 4.90 |
| FedProx [26] | **3.20** | 7.10 | 5.90 | **5.65** | 4.85 | 5.20 | 1.60 | **5.80** | 4.92 |
| SCAFFOLD [27] | 2.65 | 7.75 | **6.30** | 5.35 | 5.00 | **6.35** | 1.45 | 4.90 | **4.97** |
| FedAvgM [31] | 3.00 | 7.80 | 5.35 | 5.00 | 5.30 | 5.65 | 1.90 | 5.00 | 4.86 |
| FedYogi [32] | 2.00 | **8.45** | 6.15 | 4.55 | 3.85 | 6.30 | 1.65 | 4.93 | 4.73 |
| FedAdagrad [32] | 2.50 | 7.85 | 5.15 | 5.25 | 4.45 | 5.75 | 1.55 | 5.50 | 4.75 |
| FedAdam [32] | 2.40 | 8.50 | 5.25 | 4.70 | 4.35 | 5.40 | 1.90 | 5.20 | 4.71 |

Table 3: Experiments on single-turn chat dataset Fed-ChatbotIT. FL methods perform consistently better under open-ended instruction-following evaluations and comparably under closed-ended knowledge evaluations compared to local training. Overall, FedAdagrad is the most effective.

| Algorithm | MT-Bench-1 | Vicuna | Ref-GPT4 | Average | HumanEval | MMLU |
|---|---|---|---|---|---|---|
| Local Training | 3.73 | 6.78 | 4.49 | 5.00 | 13.41 | 46.31 |
| FedAvg [7] | 4.30 | 6.93 | **5.29** | **5.51** | 14.02 | 46.10 |
| FedProx [26] | 4.25 | 7.21 | 5.00 | 5.49 | 14.63 | 46.12 |
| SCAFFOLD [27] | 3.86 | 7.35 | 4.82 | 5.34 | 15.24 | 46.02 |
| FedAvgM [31] | **4.34** | 7.17 | 4.76 | 5.42 | 14.63 | 46.13 |
| FedYogi [32] | 4.13 | 7.20 | 5.00 | 5.44 | **15.85** | 46.24 |
| FedAdagrad [32] | 3.94 | **7.50** | 4.99 | 5.48 | **15.85** | **46.48** |
| FedAdam [32] | 3.88 | 7.32 | 5.02 | 5.41 | 14.57 | 46.10 |

comprehensive superiority in all languages, implying the necessity of future exploration of language personalization [68, 69]. (3) FedAvg and FedProx are the two most effective algorithms here.

**Fed-ChatbotIT.** Here, we conduct experiments on Fed-ChatbotIT evaluated under 5 metrics. We randomly sample two clients to run local training and average their evaluation results. From Table 3, we see that (1) on open-ended evaluation, all FL methods consistently outperform local training, indicating the effectiveness of FL in enhancing the capability of instruction following. (2) On closed-ended evaluation, FL methods perform better or are comparable to local training, indicating that FL training will not compromise LLMs' general capability.

**Fed-WildChat.** Here, we show two series of experiments based on Fed-WildChat: instruction tuning based on single-turn and multi-turn conversations, in Table 4. For both experiments, we see that FL methods consistently outperform local training, verifying the effectiveness of collaboration. For single-turn, we see that no FL method can dominate in all evaluation metrics; while for multi-turn, we see that FedAvg [7] consistently outperforms the best across metrics.

This is an interesting observation since the other baseline methods are shown to be effective in tackling data heterogeneity in other tasks such as image classification [26, 27]. This phenomenon could be attributed to two reasons: (1) training from pre-trained model itself benefits tackling the issue of data heterogeneity [70, 71], which could make some model-level optimization techniques not as effective as before [26, 27]. (2) We are fine-tuning with parameter-efficient fine-tuning technique [50] with a small number of local steps (e.g., 10), reducing the risk of overfitting on local

Table 4: Experiments of single-turn and multi-turn chat on Fed-Wildchat. FL methods perform consistently better than local training. FedAvg is a robust method in this scenario.

| Experiment Algorithm | Single-Turn | | | Multi-Turn | | | |
|---|---|---|---|---|---|---|---|
| | MT-1 | Vicuna | Ref-GPT4 | MT-1 | MT-2 | MT-Bench | Ref-GPT4 |
| Local Training | 4.15 | 7.03 | 4.50 | 3.99 | 2.56 | 3.27 | 4.68 |
| FedAvg [7] | 4.81 | 7.99 | 5.88 | **4.84** | **3.15** | **3.99** | **5.86** |
| FedProx [26] | **4.86** | 7.93 | 5.74 | 4.58 | 2.92 | 3.75 | 5.26 |
| SCAFFOLD [27] | 4.78 | 7.93 | 5.57 | 4.46 | 3.13 | 3.79 | 5.25 |
| FedAvgM [31] | 4.52 | **8.07** | 5.85 | 4.53 | 2.77 | 3.65 | 5.34 |
| FedYogi [32] | 4.78 | 8.04 | 5.48 | 4.59 | 2.96 | 3.78 | 5.05 |
| FedAdagrad [32] | 4.76 | 7.76 | **5.93** | 4.64 | 3.03 | 3.84 | 5.17 |
| FedAdam [32] | 4.54 | 8.03 | 5.68 | 4.63 | 2.85 | 3.74 | 4.96 |

Table 6: Experiments of federated preference alignment on Fed-ChatbotPA dataset. FL methods consistently perform better than local training, indicating the significance of collaboration via FL. Compared to base model, models trained via FL methods achieve consistent improvement in instruction-following capabilities and safety, and preserve most of the knowledge.

| Algorithm | MT-Bench-1 | Vicuna | Average | AdvBench | MMLU |
|---|---|---|---|---|---|
| Base Model | 3.96 | 6.31 | 5.14 | 9.40 | **40.41** |
| Local Training | 4.12 | 6.62 | 5.37 | 11.0 | 38.26 |
| FedAvg [7] | 4.44 | 7.06 | 5.75 | **16.2** | 39.70 |
| FedProx [26] | 4.44 | **7.11** | 5.78 | 13.8 | 39.51 |
| SCAFFOLD [27] | 4.53 | 7.01 | 5.77 | 16.0 | 39.94 |
| FedAvgM [31] | **4.71** | 6.87 | **5.79** | 13.3 | 39.78 |
| FedYogi [32] | 4.33 | 6.62 | 5.48 | 11.3 | 40.27 |
| FedAdagrad [32] | 4.40 | 6.79 | 5.60 | 11.0 | 40.30 |
| FedAdam [32] | 4.31 | 6.72 | 5.52 | 11.8 | 40.26 |

datasets [27, 52]. Therefore, we call for more future works to enhance the performance regarding data, such as considering data quality [12] or synthetic data [21].

**Fed-ChatbotPA.** Here, we conduct experiments of federated preference alignment on Fed-ChatbotPA dataset, with an instruction-tuned LLM as the model initialization. We randomly sample two clients to run local training and average their evaluation results. From Table 6, we see that (1) preference alignment could enhance the LLMs' capability in following humans instructions in an helpful and harmless manner. (2) All FL methods consistently perform better than local training, indicating the effectiveness of

Table 5: Results on unfiltered Fed-WildChat. FL methods consistently and evidently outperform local training.

| Method | MT-1 | Vicuna | Ref-GPT4 | Avg |
|---|---|---|---|---|
| Local | 4.15 | 7.03 | 4.50 | 5.23 |
| FedAvg | 4.61 | 8.03 | 5.81 | 6.15 |
| FedProx | **4.69** | 7.98 | 5.98 | 6.22 |
| Scaffold | 4.48 | 7.95 | 5.83 | 6.09 |
| FedAvgM | 4.63 | **8.24** | 5.99 | 6.29 |
| FedYogi | 4.68 | 7.97 | 5.47 | 6.04 |
| FedAdagrad | 4.46 | 7.98 | 5.55 | 6.00 |
| FedAdam | 4.68 | 8.11 | **6.16** | **6.32** |

federated preference alignment. Since the high-quality preference data usually involves massive human efforts, each party is hard to scale up the data, motivating diverse parties to collaborate via FL [48, 72, 73]. (3) Regarding instruction-following capabilities, FedAvgM [31], FedProx [26], SCAFFOLD [27], adn FedAvg [7] are four most effective methods.

**Results on unfiltered Fed-WildChat**. Here, we conduct instruction tuning based on the unfiltered version of Fed-WildChat (single-turn conversations). Due to the increased number of total clients, we accordingly increase the number of clients participating per round and the total communication rounds; see details in Table 11. From Table 5, we can see that FL methods consistently and evidently

Table 7: Experiments of exploration of efficient collaboration among languages. FedSimLang performs better than FedSamLang on some languages, indicating its partial effectiveness and calling for future works on constructing efficient collaboration structure to facilitate multilingual collaboration.

| Algorithm | ar | es | en | fr | pt | ru | zh | Average |
|---|---|---|---|---|---|---|---|---|
| Local | 2.55 | 5.55 | 7.20 | 4.20 | 3.85 | 4.50 | 4.95 | 4.69 |
| FedAvg | 2.50 | 5.50 | **8.00** | 5.35 | **4.95** | **5.65** | 5.25 | 5.31 |
| FedSamLang | **3.30** | **5.90** | 7.65 | **6.45** | 4.10 | 4.80 | 5.35 | **5.36** |
| FedSimLang | 3.05 | 5.85 | 7.80 | 5.40 | 4.90 | 4.30 | **5.75** | 5.30 |

outperform local training, indicating the benefits of joining collaboration. On average, we see that FedAdam [32] performs the best across baseline methods.

## 4.2 Further explorations

**Multilingual collaboration.** We have observed in Table 2 that despite that FL methods achieve better performance than local training on average, they fail to bring consistent benefits on every specific language. Such observation motivates us to explore language personalization. Therefore, in this experiment, we construct two representative baselines: FedAvg among clients with the same language (FedSamLang) and FedAvg among clients with "similar" languages (FedSimLang) to explore the potential mutual benefits among languages. We partition languages into five "similar" groups by their language family [74] as follows: (1) Standard Arabic, Urdu, and Iranian Persian (evaluated on "ar" testset); (2) French, Italian, Spanish, and Portuguese (evaluated on "fr", "es" and "pt" testset); (3) English and German (evaluated on "en" testset); (4) Russian, Polish and Ukrainian (evaluated on "ru" testset); (5) Simplified Chinese, Traditional Chinese, Japanese and Korean (evaluated on "zh" testset). Also, FedSamLang and FedSimLang generally have fewer training samples than FedAvg as they involve only one or a few of these languages.

We show the experimental results in Table 7, where we compare FedSamLang and FedSimLang with FedAvg (trained on 8 languages as previous experiments) and local training. From the results, we can see that (1) FedSamLang outperforms local training in all languages and achieves the highest average score, indicating the benefits of collaboration among clients with the same language. (2) Compared to FedSamLang, FedSimLang performs better on 3 languages (i.e., en, pt, and zh) but worse on other languages, showing that leveraging the power of other languages can benefit some particular languages. Though this observation verifies the possibility of multilingual collaboration, we need more future works to fully explore its potential. (3) FedSamLang and FedSimLang perform better or comparably compared to FedAvg with fewer collaborators, indicating the effectiveness of language personalization. These results call for future works on exploring personalization techniques that can strike a good balance between localization and collaboration or construct a better collaboration structure among these multilingual clients [69].

**Differential privacy.** Here, we conduct experiments to evaluate the effectiveness of differential privacy [75], where we apply user-level differential privacy [76]. Experiments are conducted on our Fed-WildChat single-turn dataset. We use a batch size of 1 for this experiment; see more details in Appendix D. We fix the $\delta = 1e^{-4}$ and tune $\epsilon$ in range of $\{1e^{-3}, 1e^{-2}, 0.1, 1\}$ that satisfies $(\epsilon, \delta)$-differential privacy, and report the results in Figure 6. Results show that (1) FedAvg with $(1, 1e^{-4})$-differential privacy can achieve comparable performance compared to FedAvg without differential privacy. (2) With the reduction of $\epsilon$, the privacy preservation improves while the performance degrades. FedAvg with $(1e^{-3}, 1e^{-4})$-differential privacy can achieve comparable performance compared to local training without differential privacy technique. To the best of our knowledge, this is the first time in the literature demonstrating the results of differential privacy in FedLLM. We also encourage future works to explore sample-level differential privacy [77].

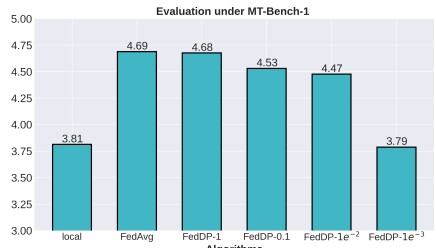

Figure 6: Experiments on Fed-WildChat with differential privacy $(\epsilon, \delta)$ $(\delta = 1e^{-4})$. FedDP-x indicates $\epsilon = x$. FedAvg with $(1e^{-2}, 1e^{-4})$-DP still outperforms local training without DP while ensuring user-level differential privacy.

**Experiments with other LLM backbone: Mistral-7B.** Previous experiments on instruction tuning task are conducted on Llama-2-7B [34]. Here we conduct experiments on Fed-WildChat using Mistral-7B [35] as the base LLM to validate the generalization ability of our benchmark across different LLM backbones. From Table 8, we can see (1) clear gap between federated learning methods and local training, (2) improvement in all metrics compared to LLaMA-2-7B results in Table 4. Notably, this increase is uniform across all baselines, and is clearly attributed to the increased capacity of the model. This

Table 8: Results based on Mistral-7B. Federated learning methods consistently outperform local training, showing the generalization ability of our benchmark across different LLM backbones.

| Method | MT-1 | Vicuna | Ref-GPT4 | Avg |
|---|---|---|---|---|
| Local | 5.23 | 8.33 | 5.47 | 6.34 |
| FedAvg | 6.08 | 8.91 | 7.21 | 7.40 |
| FedProx | 6.18 | 8.88 | 6.91 | 7.32 |
| Scaffold | 6.24 | 8.85 | 6.85 | 7.31 |
| FedAvgM | 6.20 | 8.99 | 7.14 | 7.44 |
| FedYogi | 6.31 | 9.06 | 7.00 | 7.46 |
| FedAdagrad | 6.28 | **9.11** | 6.92 | 7.44 |
| FedAdam | **6.41** | 8.88 | **7.35** | **7.55** |

indicates the evident benefits brought by collaboration and the generalization ability of our benchmark across different LLM backbones.

## 5 Conclusions

Federated learning enables multiple parties to collaboratively train large language models without sharing their raw data (FedLLM), which has attracted many research efforts from the community. In this paper, we propose the first realistic benchmark for FedLLM, FedLLM-Bench, which involves 8 training methods, 4 training datasets, and 6 evaluation metrics. The core contribution lies in the datasets, which cover a wide range of client scale and two common tasks (i.e., instruction tuning and preference alignment). These datasets exhibit many real-world diversities, including language, quality, quantity, instruction, sequence length, embedding, and preference, mirroring real-world scenarios. Based on FedLLM-Bench, we conduct extensive experiments on all datasets to benchmark classical FL methods. Besides, we also conduct experiments to explore the effective collaboration strategies of cross-language collaboration and show results when incorporating differential privacy with federated instruction tuning. We believe that our FedLLM-Bench could benefit the FedLLM community by reducing required efforts, providing a practical testbed, and promoting fair comparisons.

## Acknowledgments

This research is supported by the National Key R&D Program of China under Grant 2021ZD0112801, NSFC under Grant 62171276 and the Science and Technology Commission of Shanghai Municipal under Grant 21511100900 and 22DZ2229005.

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

Table 9: Results of perplexity on our four datasets. Higher perplexity observed on Fed-Aya and Fed-WildChat due to their high quality.

| Dataset | Fed-Aya | Fed-ChatbotIT | Fed-WildChat | Fed-ChatbotPA |
|---|---|---|---|---|
| Perplexity | 3.941 | 2.881 | 3.395 | 2.868 |

## A  Limitations

Firstly, we explore Llama-2-7B [34] and Mistral-7B [35] for instruction tuning task and Alpaca [67] for preference alignment task. More future works are required to explore more model series and sizes. Secondly, safety alignment is also an important topic in the era of LLMs, which is not comprehensively covered in our paper. This could be an interesting and promising future direction.

## B  Datasets

### B.1  Lengths measurement

To measure each data sample, We use Llama2 tokenizer to tokenize the instruction and response of each data sample and use the number of tokens as the sentence length. Figure 1(b) shows the distribution of length of instruction and response of clients' data of our four datasets.

### B.2  Verbs and nouns

We show the top 20 verbs and corresponding top 4 nouns of instructions of overall four datasets in Figure 7. We refer to the visualization code of Self-instruct [57]. Note that for all four datasets, we choose clients with English samples. From Figure 7, we can observe that different datasets possess diverse instruction types and distributions. For example, the top 2 verbs for Aya and WildChat datasets are (explain, write) and (write, make), respectively; While ChatbotIT and ChatbotPA, which are from the same public dataset, have a large range of keyword overlap but different quantity distributions.

We also show the top 20 verbs and corresponding top 4 nouns of instructions of individual clients from four datasets in Figure 9, Figure 10, Figure 11 and Figure 12. For all four datasets, we choose clients with English samples to present verbs and nouns in their instructions.

### B.3  Quality evaluation

We conduct data quality evaluation with the pre-trained Llama2-7B [34] and the IFD metric [59]. IFD is a quality evaluation metric, qualifying the instruction-following difficulty of the given model as the data quality. It has been widely used in [78, 21, 79]. From Figure 3, we can see that clients in four datasets have various data qualities. This indicates these four federated datasets demonstrate quality heterogeneity, which is an inherent property of real data sets.

### B.4  t-SNE visualization

We also implement the t-SNE instruction-response embedding in four datasets. Here the *'text-embedding-ada-002'* from OpenAI is utilized as the feature extraction model. We randomly select 10 clients from each dataset and use the t-SNE two-dimensional visualization to demonstrate the data heterogeneity from each client. From Figure 4, we could see that data points from the same client cluster in the feature space. This is particularly evident in Fed-Aya and Fed-WildChat, demonstrating data heterogeneity within the dataset.

### B.5  Perplexity

Perplexity is a widely-used metric for evaluating data complexity and quality in language modeling. To provide a more comprehensive understanding of our dataset's variability and quality, We evaluate perplexity on our four datasets and the results are shown in Table 9. The model has relatively higher average perplexity on Fed-Aya and Fed-WildChat, and lower on Fed-ChatbotIT and Fed-ChatbotPA.

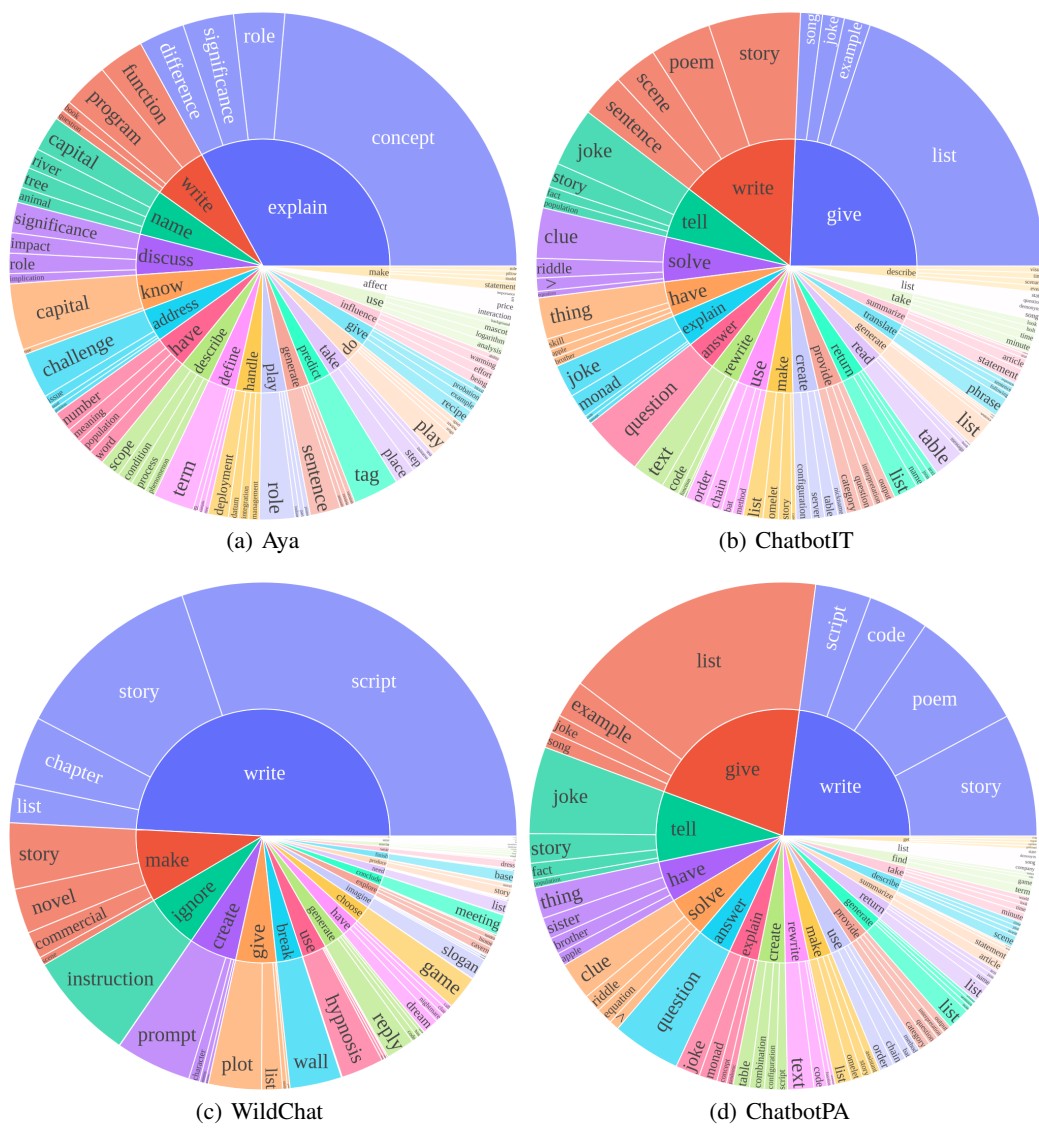

Figure 7: The top 20 most common root verbs (inner circle) and their top 4 direct noun objects (outer circle) in the instructions of four datasets.

## B.6 Unfiltered Dataset

To provide researchers with flexibility, we have released both filtered and unfiltered versions of our datasets. The unfiltered version enables future researchers to freely adjust and customize the data according to specific research needs. We also provide a overall statics of data distribution as shown in Table 10. The unfiltered dataset includes a larger number of clients, but the average data volume per client has significantly decreased, compared to the filtered version as shown in Table 1.

## B.7 Discussion about data privacy and safety

The base datasets we construct ours from have already undergone screenings for safety and privacy. For example, in Chatbot-arena Conversations dataset, most conversations that contain personally identifiable information have been moved. Fed-ChatbotIT, Fed-ChatbotPA and Fed-WildChat, which are based on Chatbot-arena Conversations dataset and WildChat dataset, may contain unsafe or toxic interacts, but they are kept so that these datasets can be better used to study AI safety and simulate real-world dialogue scenarios.

Table 10: Data distribution of unfiltered datasets. The unfiltered dataset includes a larger number of clients, but the average data volume per client has significantly decreased.

| Dataset Name | Fed-Aya | Fed-ChatbotIT | Fed-WildChat | Fed-ChatbotPA |
|---|---|---|---|---|
| # Clients (Total) | 1456 | 10996 | 181063 | 10996 |
| # Samples (Total) | 202364 | 23294 | 899215 | 23294 |
| # Samples (Client) | 139±605 | 2±6 | 5±57 | 2±6 |

Table 11: Experimental setups of all datasets. 'Local epochs' denotes the number of training epochs in local training. In the column of 'Clients', x/y denotes that there are y clients in total and we same x clients for each round.

| Dataset | Local Epochs | Clients | Local Steps | Global Rounds |
|---|---|---|---|---|
| Fed-Aya | 5 | 4/38 | 10 | 200 |
| Fed-ChatbotIT | 10 | 10/237 | 5 | 100 |
| Fed-WildChat | 5 | 5/100 | 10 | 100 |
| Fed-WildChat (Multi-Turn) | 20 | 3/50 | 10 | 50 |
| Fed-ChatbotPA | 10 | 10/747 | Dynamic | 200 |
| Fed-WildChat (unfiltered) | 5 | 10/500 | 10 | 200 |

## C   Experiments

### C.1   Experimental setups

We report the detailed experimental setups in Table 11. For each table, we randomly sample two clients to conduct local training and average their performance as the final results of 'local training' in the table. We use dynamic local steps for local training in ChatbotPA. We calculate the probability of a user being selected in a round given parameters such as the total number of rounds, the number of clients sampled per round, and the total number of clients. We then adjust the local training steps for each client based on their sampling probability and data volume, ensuring that each client's data can undergo about three epochs of training in total. Our experiments were mainlt conducted on a machine equipped with an NVIDIA GeForce RTX 3090 GPU with 24 GB of VRAM. Experiments on Fed-WildChat(Multi-Turn) were conducted on a machine equipped with an NVIDIA A40 with 48GB of VRAM.

Here, we provide a comprehensive description of the selected baselines, categorizing them into two types (classified by operations on the client or server side) to demonstrate the thoroughness of our experiments.

1. FedAvg [7] is the basic federated learning method.

2. Client-side methods: FedProx [26] applies an l2 regularization term between the local model and global model during the training of local model on the client side. SCAFFOLD [27] introduces a control variate that corrects the gradient of local model on the client side.

3. Server-side methods: FedAvgM [31] introduces simple momentum for updating the global model on the server side. FedAdagrad, FedYogi, and FedAdam [32] introduce adaptive optimization methods for updating the global model on the server side.

The hyperparameter configurations used in each baseline experiment are detailed in Table 12. To further optimize model efficacy, We perform a tuning experiment on the FedAdam baseline using the Fed-WildChat dataset, results shown in Table 13, where the adjusted parameters yield improvements. This experiment serves as a basis for future work, encouraging further exploration into optimal parameter settings that promote robust and efficient federated learning outcomes.

### C.2   Evaluation

Here, we show the prompt template used in GPT-4 Judge in Figure 8. For the specific dataset Aya, we utilize some test samples from the raw dataset, where each sample contains a question and a

Table 12: Hyper-parameters of baselines.

| Baseline | Hyper-parameters |
|---|---|
| FedProx | $\mu = 0.01$ |
| FedAvgM | momentum=0.5 |
| FedAdagrad | $\beta_1 = 0, \beta_2 = 0, \eta_g = 1e{-}3, \tau = 1e{-}3$ |
| FedYogi | $\beta_1 = 0.9, \beta_2 = 0.99, \eta_g = 1e{-}3, \tau = 1e{-}3$ |
| FedAdam | $\beta_1 = 0.9, \beta_2 = 0.99, \eta_g = 1e{-}3, \tau = 1e{-}3$ |

Table 13: Results of FedAdam on Fed-WildChat. New parameters achieve better performance.

| Method | MT-1 | Vicuna |
|---|---|---|
| FedAvgM | 4.52 | 8.07 |
| FedAdam (previous parameters) | 4.54 | 8.03 |
| FedAdam (new parameters) | 4.68 | 8.35 |

reference answer. We infer the tested model with the question and obtain an answer. Then we fill in the "question", "answer" and "reference" blanks of the template.

---

**Prompt:**

[Instruction]
Please act as an impartial judge and evaluate the quality of the response provided by an AI assistant to the user question displayed below. A good answer should follow these rules:
1.It should be in the same language as the question.
2. It should answer the request in the instruction.
3.It should be factually and semantically comprehensible.
4. It should be grammatically correct and fluent.
Begin your evaluation by providing a short explanation. Be as objective as possible. After providing your explanation, you must rate the response on a scale of 1 to 10 by strictly following this format: \"[[rating]]\", for example: \"Rating: [[5]]\". A human annotated answer is given for reference.

[Question]
{question}

[The Start of Assistant's Answer]
{answer}
[The End of Assistant's Answer]

[Reference]
{reference}

---

Figure 8: Prompt template used in GPT-4 judge.

# D   More details about differential privacy

## D.1   Definition

Differential privacy (DP) [75] has emerged as a broadly recognized framework for safeguarding privacy in statistical analyses. Through DP, we can perform computations on extensive datasets while ensuring that individual data points remain indistinguishable, thereby protecting personal privacy.

In general, we use privacy parameters $\epsilon$ and $\delta$ to formally define DP. Specifically, a randomized mechanism $M : \mathcal{D} \rightarrow \mathcal{R}$ is $(\epsilon, \delta)$-differential private for $\epsilon > 0$ and $\delta \in [0, 1)$ if for any two neighboring datasets $D, D' \in \mathcal{D}$ differing by at most one entry and for any subset of outputs $R \subseteq \mathcal{R}$ it holds that

$$\mathbb{P}(M(D) \in R) \leq \exp(\epsilon)\mathbb{P}(M(D') \in R) + \delta.$$

Table 14: MT-Bench on WildChat with Differential Privacy and fixed $\sigma$. FedLLM with DP($\sigma$=0.1) still outperforms local and differential privacy costs slight model performance while ensuring user-level differential privacy.

| local(813) | local(1702) | FedAvg | FedDP-$1e^{-3}$ | FedDP-$1e^{-2}$ | FedDP-0.1 | FedDP-1 |
|---|---|---|---|---|---|---|
| 3.5375 | 4.0875 | 4.6875 | 4.6750 | 4.5500 | 4.5375 | 1.6875 |

## D.2  User-level differential privacy

We implement user-level differential privacy(UDP) [76] in our experiments. Following [76], we use the Gaussian mechanism that employs the $L_2$ norm sensitivity. It adds zero-mean Gaussian noise with variance $\sigma^2\mathbf{I}$ to each coordinate of the function output $r(D)$ as follows:

$$\mathcal{M}(D) = r(D) + \mathcal{N}(0, \sigma^2\mathbf{I}),$$

where $\mathbf{I}$ is an identity matrix of the same size as $r(D)$. The sensitivity of the function $r$ is expressed as:

$$\Delta r = \max_{D, D' \in \mathbf{D}} \|r(D) - r(D')\|_2,$$

which provides an upper bound on the necessary perturbation to its output for privacy preservation. By appropriately selecting the value of $\sigma$, this mechanism satisfies $(\epsilon, \delta)$-differential privacy.

## D.3  Experiment details of differential privacy

In our experiments with UDP, we use WildChat dataset. For convenience, the batch size is 1 in all DP experiments and other settings are the same as single-turn WildChat experiment setting, see details in Section 4.

We add Gaussian noise cautiously controlled by $\sigma$ when local clients upload their local models to the server, ensuring User-level differential privacy. Following [76], the value of $\sigma$ is calculated by:

$$\sigma = \delta_l \frac{\sqrt{2qN \ln \frac{1}{\delta}}}{\epsilon}$$

where $q$ is the sample fraction of clients each round, $N$ is the federated learning communication round, $(\epsilon, \delta)$ is the DP parameters and $delta_l$ is decided as follows:

$$\delta_l = \frac{2\eta C}{\frac{|D|}{|n|}}$$

where $\eta$ is the learning rate, $C$ is the maximum of gradients, $|D|$ is the size of dataset and $|n|$ is the number of clients. Note that when expressed with $(\epsilon, \delta)$, smaller $\epsilon$ means smaller privacy budget, in other words, better privacy and usually lower performance. However, when expressed with $\sigma$, larger $\sigma$ means better privacy and usually lower performance.

Our experiment results are shown in Figure 6 and Table 14. Figure 6 shows that FedLLM with $(0.01, 1e^{-4})$-DP still outperforms local training and differential privacy costs slight model performance while ensuring user-level differential privacy.

In fact, the $\sigma$ calculated in Section D.2 is a proper bound when ensuring user-level differential privacy with given $(\epsilon, \delta)$. We also conduct experiments with fixed $\sigma$ and results are shown in Table 14.

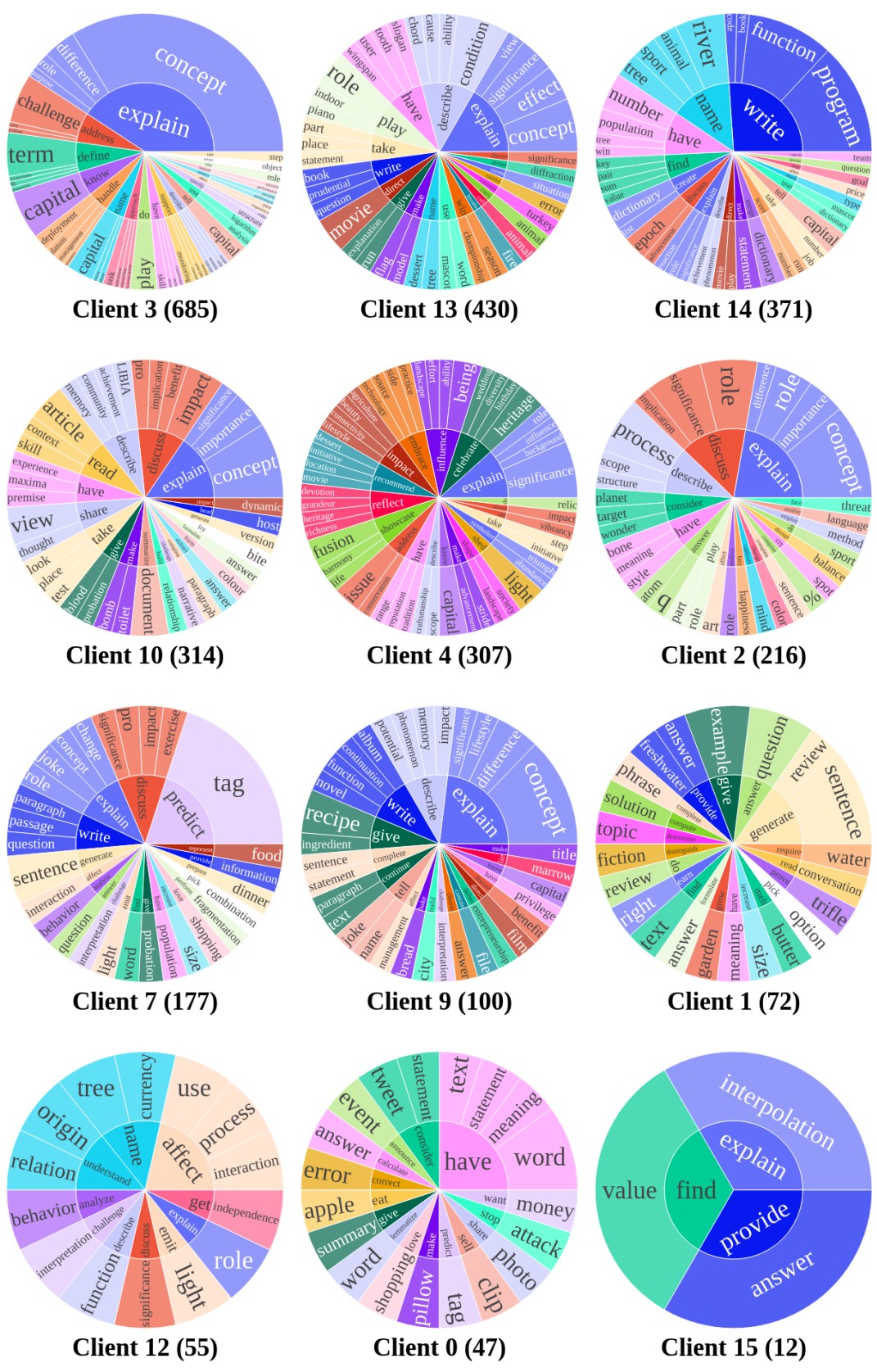

Figure 9: The top 20 most common root verbs (inner circle) and their top 4 direct noun objects (outer circle) in the instructions of **Aya** for 12 different English clients. We select English clients on purpose.

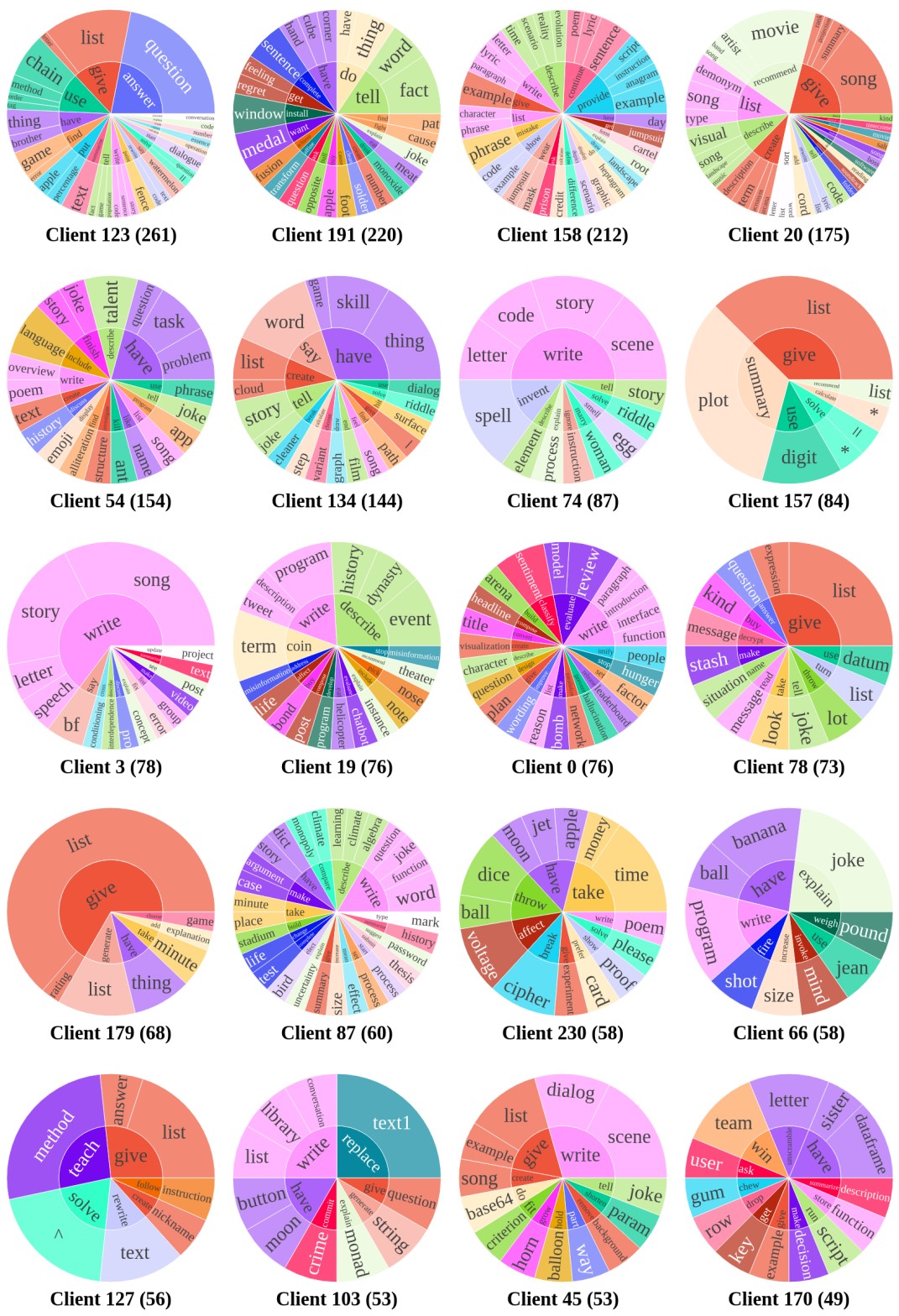

Figure 10: The top 20 most common root verbs (inner circle) and their top 4 direct noun objects (outer circle) in the instructions of **ChatbotIT** for 20 different clients.

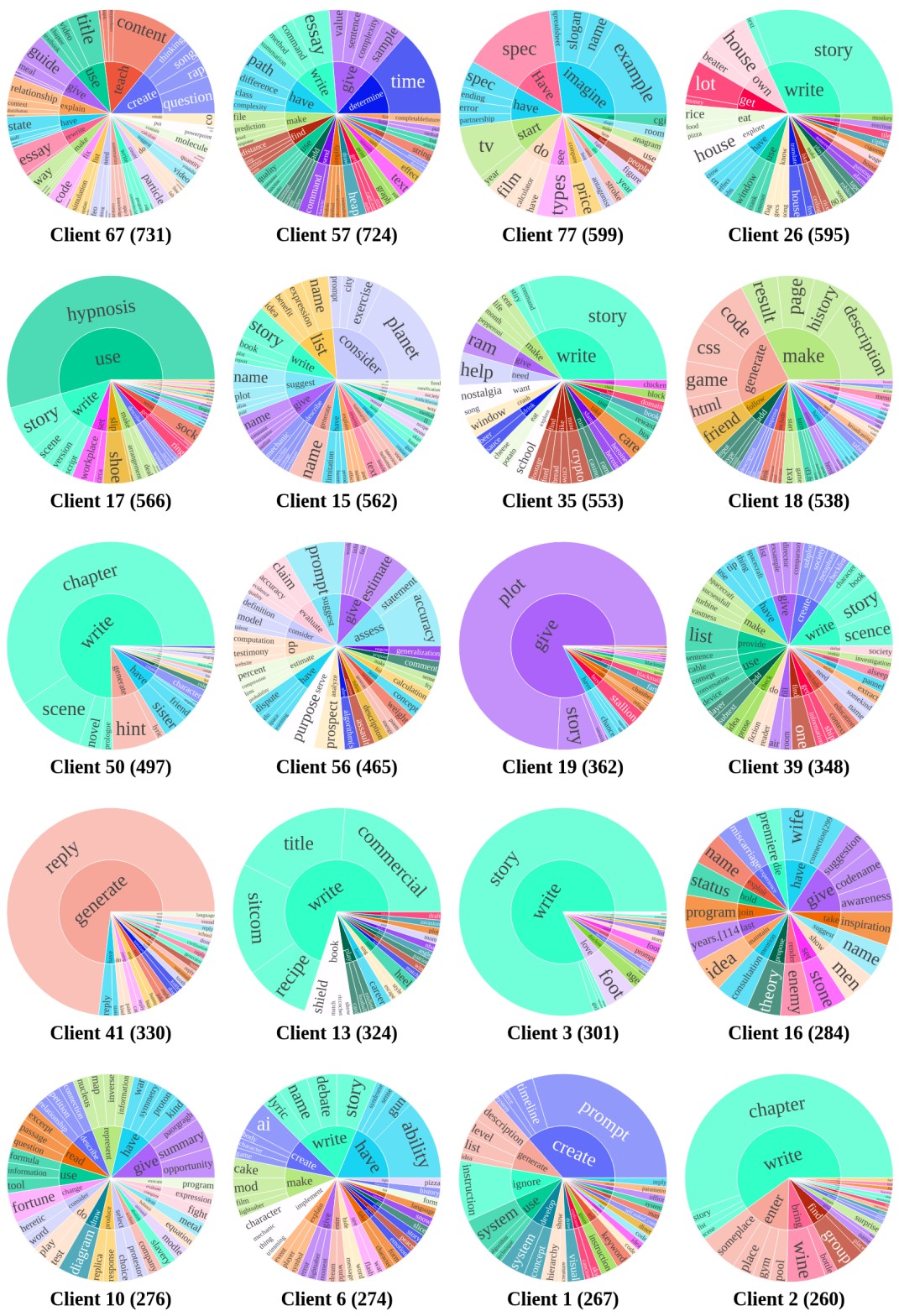

Figure 11: The top 20 most common root verbs (inner circle) and their top 4 direct noun objects (outer circle) in the instructions of **WildChat** for 20 different clients.

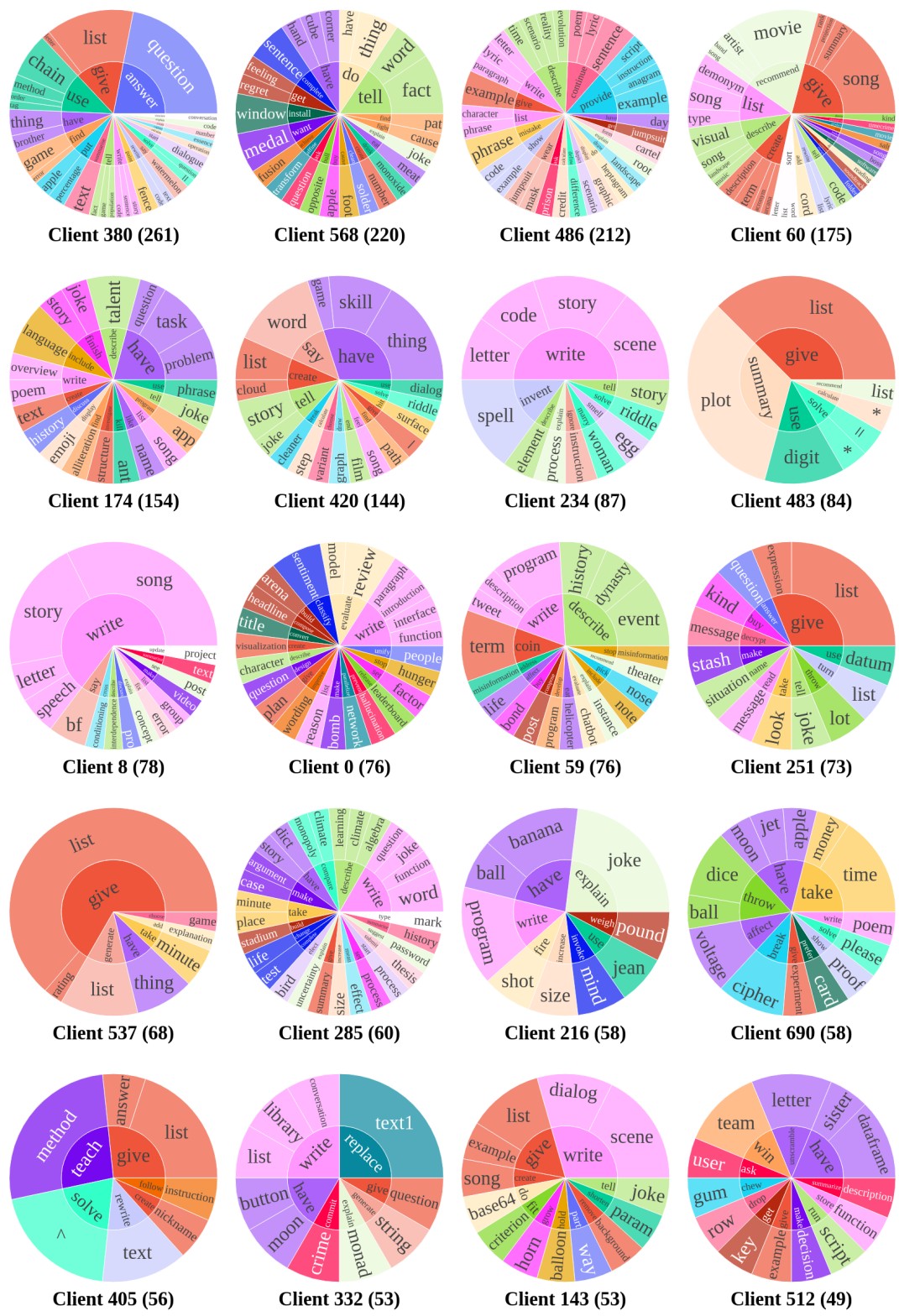

Figure 12: The top 20 most common root verbs (inner circle) and their top 4 direct noun objects (outer circle) in the instructions of **ChatbotPA** for 20 different clients.

