# FedLLM-Bench: Realistic Benchmarks for Federated Learning of Large Language Models Supplementary Materials

## 1 Dataset

### 1.1 Links and Preservation

We have provided data processing code and training benchmarks in our GitHub repository at https://github.com/rui-ye/FedLLM-Bench and the data can be downloaded at data. The croissant metadata record is available at croissant. We chose GitHub and Google Drive respectively to store our code and dataset. Both are widely recognized as reliable data storage platforms, ensuring long-term preservation. We highly recommend downloading the raw data directly and following the provided instructions to simplify the data processing steps.

### 1.2 Dataset Structure

Our dataset is structured as follows: the **local** directory contains client-specific data for local training, while **all_clients** aggregates data from all clients for federated learning. All datasets are stored in JSON format.

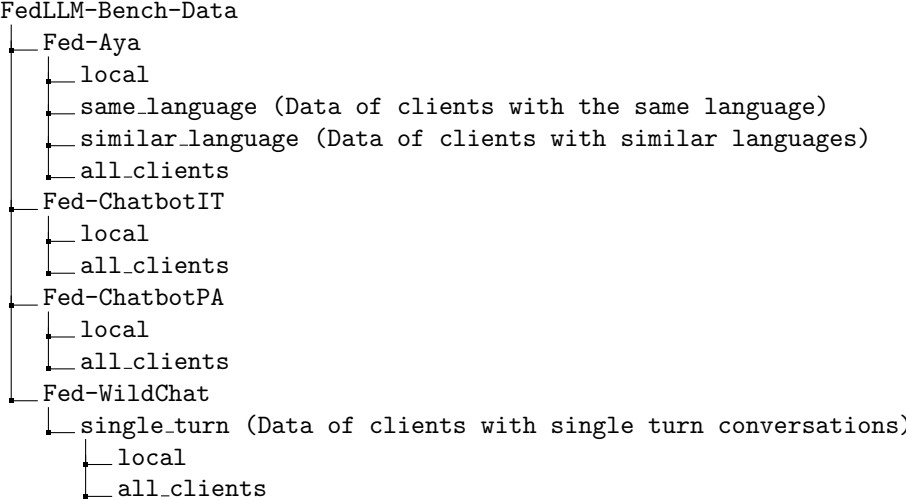

```
FedLLM-Bench-Data
├──Fed-Aya
│   ├──local
│   ├──same_language (Data of clients with the same language)
│   ├──similar_language (Data of clients with similar languages)
│   └──all_clients
├──Fed-ChatbotIT
│   ├──local
│   └──all_clients
├──Fed-ChatbotPA
│   ├──local
│   └──all_clients
└──Fed-WildChat
    └──single_turn (Data of clients with single turn conversations)
        ├──local
        └──all_clients
```

```
└─ multi_turn (Data of clients with multi-turn conversations)
   ├─ local
   └─ all_clients
```

For federated training, the data file consists of a dictionary where each key represents a user, and the corresponding value is a list containing all data samples for that user. In contrast, the data file for local training is a list of dictionaries, each representing a data sample of the user. An example data sample for instruction tuning is provided, and a similar structure applies for value alignment, where keys include **prompt**, **chosen**, and **rejected** as detailed in our publication.

```
1  {
2      "instruction":String,
3      "response":String
4  }
```

For multi_turn federated training, the data file consists of a dictionary where each key represents a user, and the value is a list of sub-dictionaries. Each sub-dictionary has two key-value pairs. The former represents the ID of the multi_turn chat of a specific user. The latter contains multi_turn chat content between human and gpt. We show a simple example as follows, a user "34131" has one multi_turn chat data, which consists of two turn conversation.

```
1  {
2      "34131": [
3          {
4              "id": "identity_0",
5              "conversations": [
6                  {
7                      "from": "human",
8                      "value": "how..."
9                  },
10                 {
11                     "from": "gpt",
12                     "value": "As..."
13                 },
14                 {
15                     "from": "human",
16                     "value": "Show..."
17                 },
18                 {
19                     "from": "gpt",
20                     "value": "Sure..." }
21             ]
22         }
23     ]
24 }
```

The pre-processing of multi_turn data is different from single_turn data. Instead of using the function *DataCollatorForCompletionOnlyLM* in *trl* library, we manually set the position to calculate loss following FastChat.

## 1.3   Intended Use

Our dataset is used to benchmark federated learning algorithms for large language models. Additionally, due to its diversity and realness, it can also be utilized for further research such as studying real user preferences and multi-language training of large language models.

## 1.4   Host and Licence

As detailed in our publication, our dataset is sourced from existing publicly available datasets. These datasets are well-known on Hugging Face and originate from reputable institutions, with original licenses already provided. Therefore, we do not provide additional licenses here.

## 1.5   Reproducibility

All code for training and evaluation is provided at https://github.com/rui-ye/FedLLM-Bench. Our results can be easily reproduced using our code.