# OpenReview forum: "FedLLM-Bench: Realistic Benchmarks for Federated Learning of Large Language Models"
_NeurIPS.cc/2024/Datasets_and_Benchmarks_Track — NeurIPS 2024 Track Datasets and Benchmarks Poster_

### Official Review · Reviewer_4gik · 2024-07-18
**A solid paper that makes misleading claims about being the first work to do a variety of things**

**Rating:** 6
**Confidence:** 5

**Review:**

In order to avoid repeating myself, I am going to list both the strengths and the weaknesses of the work here. I think it has its drawbacks in terms of **how it talks about its own contributions**, but I think the contributions that are actually there are solid.

## Strengths

In short, useful datasets for FL research really are important, and there aren't that many of them (though they do exist, and are not discussed here, see below). This paper proposes 3 datasets formed by partitioning existing datasets by interesting facets (e.g. data annotator or user).

The datasets proposed seem legitimately useful for research on FL + LLMs.  I especially appreciate the multilingual dataset, which is something that I believe is relatively understudied in FL settings.

I also think the benchmarks are good. They do a good job of trying a variety of useful methods, and better yet, they actually compare to local training. I think this is a baseline that is missing from a lot of works, and I appreciated seeing it here. I also think they do a good job of codifying the notion that FL + LLM research should strive to mirror best practices from centralized LLM research.

## Weaknesses

Put simply, the biggest weakness of the paper is that (contrary to what the authors claim) they are not the first to propose interesting, naturally heterogeneous, useful datasets for LLM + FL research.

The paper also makes a weird distinction between other works that "artificially partition centralized datasets" and their own datasets (which are partitioned versions of centralized datasets, in ways that are similar to other datasets in the literature).

Another issue is that the datasets are relatively small. For all 3, the authors restricted to those FL clients that only had more than $X$ samples (where $X$ depends on the dataset). I don't understand why they did so - this just makes the dataset less heterogeneous, and less useful for cross-device FL research. I really think the datasets would be much more useful if used in full. Why not pass $X$ into the training, so that users of the benchmark can decide how much to filter to, instead of forcing it? This is actually really important - it makes the datasets smaller, and less representative of cross-device FL, in which some clients may have only a single example.

The authors focused on breadth of algorithmic comparison (e.g. FedAdagrad & FedYogi & FedAdam, instead of just picking FedAdam) over doing more nuanced tuning. As a consequence, the results are noisy and seemingly uncorrelated. In Table 3, I would've expected FedAdam, when properly tuned, to always outperform FedAdagrad, and perform at least as well as FedAvgM (based on [Reddi et al., 2021]). I suspect that large differences between similar methods are more to do with the lack of tuning than anything else.

**Strengths:**

Please see **Strengths** above.

**Additional Feedback:**

I have marked the paper as borderline only because I don't think the paper should be accepted **in its current state.** I fully believe the authors can remedy the issues I have outlined above (especially the discussion of the novelty of this work related to other datasets for FL + LLMs) and should.

**Clarity:**

Generally yes. I appreciated the clarity of experimental setup and evaluation especially.

**Correctness:**

The experiments seem correct - with the caveat that they are not particularly well-tuned.

**Documentation:**

The authors really don't talk about hosting plans and licensing. While the datasets are available on drive now, the supplement does not discuss future plans.

The license is also concerning, the authors state (verbatim):
```
As detailed in our publication, our dataset is sourced from existing publicly available datasets. These datasets are well-known on Hugging Face and originate from reputable institutions, with original licenses already provided. Therefore, we do not provide additional licenses here.
```
I'm not an expert on dataset licensing, but this seems short on details. I think it would be much better for the authors to discuss the dataset-specific licenses, and hopefully justify that their hosting of the dataset falls under the licenses.

**Ethics:**

None.

**Limitations:**

This part seems fine.

**Opportunities For Improvement:**

Please see **Weaknesses** above. But in summary, I believe the authors should address the following:

1. How their datasets fit into the landscape of realistically heterogeneous datasets for FL + LLM research.
1. Why their datasets that partition existing centralized datasets are different from other works that do this. I think the key here is dataset-agnostic partitioning, versus dataset-specific partitioning (which is done in other work, but is not mentioned here).
1. Why do you cap the datasets to only users with a certain number of examples? Can the benchmark be altered to instead allow users to experiment with changing the cap themselves, so that if a user wants, they can actually use the entire dataset?

**Relation To Prior Work:**

This is probably the weakest aspect of the work. Put frankly, the work does not engage seriously with a large amount of work that attempts to do good federated learning + LLM research, including benchmarking and evaluation. I am sympathetic to the authors in the sense that there is a deluge of work on this front right now, but this needs to be addressed before I can recommend acceptance of the paper.

In short, there are lots of non-artificially constructed datasets used for research on FL & LLMs already in existence. Perhaps most relevant is [1], which was (a) published at last year's NeurIPS datasets & benchmarks track, (b) specifically designed for foundation model training, and (c) actually benchmarked on significantly larger naturally heterogeneous datasets. However, there are other works that use naturally heterogeneous language datasets for research on federated learning & LLMs, see [2, 5] for example.

This is not to diminish from the contributions of this work (which I think are good). However, the paper is essentially incorrect in many of its statements about being the first to propose useful, naturally heterogeneous datasets for FL + LLM research:

* L4-L7: "However, an unpleasant fact is that there are  currently no realistic datasets and benchmarks for FedLLM and previous works all rely on artificially constructed datasets, failing to capture properties in real-world scenarios."
* L32-33: "there is currently no realistic benchmark for FedLLM..."
* L34-35: "each previous work constructs its own FL datasets by artificially partitioning existing centralized datasets"
* L39: "To fill this gap, we propose the first realistic benchmark for FedLLM..."
* L66: "We propose the first realistic FedLLM benchmark..."
* L88: "Addressing this, we propose the first realistic benchmark for FedLLM, FedLLM-Bench..."
* L98-L99: "However, currently, there is no realistic dataset or benchmark for the tasks of FedLLM, while our FedLLM-Bench stands out as the first one in the literature..."
* L273: "we propose the first realistic benchmark for FedLLM..."

### Artificial versus Natural Partitioning of Datasets

On a related note, I think this work misrepresents what "artificially constructed" means. My comparison point is going to be [3], which creates datasets for FL + LLMs via things like taking a code-based dataset, and partitioning based on programming language. The authors of this work seem to say that this is just "artificially partitioning existing centralized datasets". However, I would urge the authors to consider their own datasets. Take Fed-Aya. Aya is an **existing centralized dataset**. In this work, the authors propose partitioning it by data annotator. Why is this any more or less artificial than partitioning by programming language? In my opinion, both approaches take a useful facet of the dataset, and partition according to it (rather than doing **dataset-agnostic partitioning**). But it is not clear to me that one is inherently more or less useful than the other.

### Existing Widely-Used Datasets

One last note I'll make is that there are existing FL datasets that are already useful for FL + LLM research. The Stack Overflow dataset, which has been used for years (since at least 2020) was used for FL + LLM research in [4] to good effect. The dataset has natural heterogeneity, a large number of users and examples, and a variety of other types of heterogeneity (e.g. variance in examples per user). It is not clear to me why this is not referenced in the work. If it is about sequence length and how appropriate this is for LLMs, that's fine, but it should be referenced. I note that the suitability of Stack Overflow and other popular existing FL datasets was discussed in depth in [1].

### References

1. Charles et al., "Towards Federated Foundation Models: Scalable Dataset Pipelines for Group-Structured Learning." NeurIPS Datasets & Benchmarks, 2023.
1. Collins et al., " Profit: Benchmarking Personalization and Robustness Trade-off in Federated Prompt Tuning."
1. Kuang et al., "FederatedScope-LLM: A Comprehensive Package for Fine-Tuning Large Language Models in Federated Learning."
1. Wang et al., "Can Public Large Language Models Help Private Cross-device Federated Learning?"
1. Cho et al., "Heterogeneous LoRA for Federated Fine-tuning of On-Device Foundation Models."

**Summary And Contributions:**

At its core, this paper provides 4 federated datasets that could be used for benchmarking federated learning methods on (large) language models. These datasets have naturally occurring heterogeneity (e.g. often partitioned by data annotator) and encapsulate some of the interesting language-specific challenges to language model training. The authors also do some benchmarking of federated LoRA fine-tuning on these datasets, generally comparing a variety of federated optimization methods to purely local training.

Something I want to call out here though that the something that is not a contribution of the paper (that the authors tout repeatedly) is that they are the first to provide any realistic language modelling datasets for federated learning of LLMs. This has actually done in a variety of works already in the literature. I will discuss this in the review below, but I want to bring it up here as it represents the biggest difference between the stated contributions of the work and what is actually attained.

---

> ### Author Rebuttal · Authors · 2024-08-16
>
> First of all, we sincerely thank the reviewer for giving so many comments in detail, which greatly help us to improve our work. Addressing your concerns and following your suggestions, we have the following detailed responses.
>
> ---
>
> **W1:** Put simply, the biggest weakness of the paper is that (contrary to what the authors claim) they are not the first to propose interesting, naturally heterogeneous, useful datasets for LLM + FL research.
>
> **Response:** Thanks for pointing out this confusion and sorry for that. In our paper, the term FedLLM-Bench actually refers to federated fine-tuning of LLMs, which covers two of the typical steps in fine-tuning LLMs (instruction tuning and preference alignment), excluding the step of pre-training. Here, as well as in the revision, we will further clarify our claim as follows: our paper is the first to propose naturally heterogeneous datasets and benchmarks for federated **fine-tuning** of large language models (especially in the post-ChatGPT era); while we acknowledge that there are several existing realistic datasets for federated **pre-training** of LLMs. Here, 'naturally heterogeneous' means that the datasets are split by real-world user ID (e.g., individual or company).
>
> **Why we focus on federated fine-tuning of LLMs rather than pre-training**.
> It is commonly concluded that three key steps for training chatbot-level LLMs are: pre-training, instruction tuning, and preference alignment. Among these, pre-training often requires high compute resource, massive general public corpus, and is inapplicable with parameter-efficient tuning techniques such as LoRA, making it much more challenging to be achieved via federated learning (at least for now). Therefore, we focus on **instruction tuning and preference alignment**, which can be achieved by moderate amounts of specific data samples, better fitting the cross-user setups in federated learning. This would make it easier for researchers without supercomputing resources to participate in Fed+LLM research, helping to promote related studies.
>
> **Our unique contributions**
> 1. **[Different task]** While we acknowledge that there are existing (naturally heterogeneous) datasets for language modeling in FL, they are all (or almost) for the process of pre-training, such as Shakespeare [6], Stack Overflow [7] In contrast, our datasets and benchmarks focus on instruction tuning or preference alignment, which are unique contributions to the FL community.
> 2. **[Different data format]** Further, the data format of our datasets are different from pre-training datasets in existing works. For the instruction tuning task, each data sample consists of an instruction and a corresponding response; for the preference alignment, each data sample typically consists of an instruction and two responses (preferred and dispreferred. Both of them appear in a structured manner, which might require massive human annotations. In contrast, for the pre-training task, each data sample is a sequence of text, which appears in an unstructured manner (e.g., any raw text from public web might become one data sample).
>
> We will clearly state this part in our revision and also include more related works.
>
> [6] McMahan, Brendan, et al. "Communication-efficient learning of deep networks from decentralized data." Artificial intelligence and statistics. PMLR, 2017.
>
> [7] https://www.tensorflow.org/federated/api_docs/python/tff/simulation/datasets/stackoverflow/load_data
>
> ---
>
> **W2:** The paper also makes a weird distinction between other works that "artificially partition centralized datasets" and their own datasets (which are partitioned versions of centralized datasets, in ways that are similar to other datasets in the literature).
>
> **Response:** Sorry for the confusion. In this paper, we define datasets as 'naturally partitioned' if they are split by real-world user ID (e.g., individual or company). Therefore, the key difference between 'artificially partitioned' (e.g., random partition or partition based on data labels) and 'naturally partitioned' lies in whether they are split by real-world user ID. Please note that **such definition follows the statements in several previous benchmark papers**, including FLAIR [8] (partitioned by user ID of Flicker) and FLamby [9] (partitioned by user ID, i.e., hospital ID), Sentiment140 [10] (partitioned by user ID of Twitter). Note that 1) both [8] and [9] are accepted by NeurIPS dataset track 2022; and 2) similar to our datasets, [8,9,10] are also partitioned from centralized datasets via real-world user IDs. Following the paradigm in [8,9,10], since all of our datasets are also partitioned by real-world user IDs, we thus call them 'naturally partitioned' realistic datasets. We will clarify this point in the revision.
>
> [8] Song, Congzheng, Filip Granqvist, and Kunal Talwar. "Flair: Federated learning annotated image repository." Advances in Neural Information Processing Systems 35 (2022): 37792-37805.
>
> [9] Ogier du Terrail, Jean, et al. "Flamby: Datasets and benchmarks for cross-silo federated learning in realistic healthcare settings." Advances in Neural Information Processing Systems 35 (2022): 5315-5334.
>
> [10] Caldas, Sebastian, et al. "Leaf: A benchmark for federated settings." arXiv preprint arXiv:1812.01097 (2018).

---

> > ### Author Rebuttal · Authors · 2024-08-16
> >
> > **W3:** Another issue is that the datasets are relatively small. ... This is actually really important - it makes the datasets smaller, and less representative of cross-device FL, in which some clients may have only a single example.
> >
> > **Response:**
> >
> > Thanks for the advice. To address your concerns, we will respond from two perspectives: our new experimental results and the reasons why we chose the scale in our paper.
> >
> > &nbsp;
> >
> > **First**, we have now released the unfiltered version of each dataset (see the statistics in Table R1) and conduct initial experiments on top of it. Due to limited time, here, we sample 300 clients of Fed-WildChat for the experiments, where some of the clients hold only a single sample. From Table R2, we still see that federated learning methods consistently and evidently outperform local training, indicating the effectiveness of joining collaboration.
> >
> > [Table R1. Data distribution of unfiltered dataset]
> > | Dataset Name | Fed-Aya | Fed-ChatbotIT | Fed-WildChat | Fed-ChatbotPA |
> > |:--------:|:-----:|:---------:|:--------:|:--------:|
> > |# Clients (Total)|1456|10996|181063|10996|
> > |# Samples (Total)|202364|23294|899215|23294|
> > |# Samples (Client)|139±605|2±6|5±57|2±6|
> >
> > &nbsp;
> >
> > [**Table R2.** Results on unfiltered Fed-WildChat.]
> > | Method | MT-1 | Vicuna | Ref-GPT4 |
> > |:--------:|:-----:|:---------:|:--------:|
> > |Local|4.15|7.03|4.50|
> > |FedAvg|4.61|8.03|5.81|
> > |FedProx|4.69|7.98|5.98|
> > |Scaffold|4.48|7.95|5.83|
> > |FedAvgM|4.63|8.24|5.99|
> > |FedYogi|4.68|7.97|5.47|
> > |FedAdagrad|4.46|7.98|5.55|
> > |FedAdam|4.68|8.11|6.16|
> >
> > &nbsp;
> >
> > **Second**, in the following, we would like to explain why we choose the current scale, and we hope to gain your understanding. The reasons are two-folded.
> >
> > - Such operation is inspired by experience from centralized training of LLMs that iterating too many times on training samples could affect the performance. In centralized training of LLMs [11], the number of epochs is often set as a small number (e.g., 3) and the batch size is often set as a moderate number (e.g., 16, 32, 64, 128). Training too much on data samples or using a small batch size could lead to bad performance. Therefore, in our federated learning setting, we need to consider this similar issue. Let's assume that we set the number of local training steps as 10 and a batch size 16. Then, for a client with few training samples (e.g., only 1). At each round, the data of this client will be iterated for 10 epochs, which is improper according to experience from centralized training. Therefore, we choose to filter out clients with too few samples.
> >
> > - Choosing such a moderate scale makes the training affordable for a wide range of researchers, which promotes the development of this field by avoiding it becoming the game of only a few players (e.g., large companies). Without filtering, the number of clients could be over 10,000, indicating that we need to run 10,000+ large language models (or their adapters) on their compute resources, which is unaffordable for many researchers.
> >
> > &nbsp;
> >
> > Overall, we believe that the filtered version would contribute to involving more researchers in this field. Meanwhile, we agree that the unfiltered version could also play its role, which has been released following your suggestions.
> >
> > [11] https://github.com/lm-sys/FastChat/tree/main
> >
> > ---
> >
> > **W4:** The authors focused on breadth of algorithmic comparison (e.g. FedAdagrad & FedYogi & FedAdam, instead of just picking FedAdam) over doing more nuanced tuning. ... I suspect that large differences between similar methods are more to do with the lack of tuning than anything else.
> >
> > **Response:**
> >
> > We would like to respond from three perspectives.
> >
> > **First**, for FedAdagrad, FedYogi, and FedAdam, we have tuned the hyper-paramter of global learning rate (\eta_g) and $\tau$ and report the best results. While we agree that a more comprehensive grid search could potentially lead to better performance, we need the attention of the reviewer that this could involve hundreds of experiments for each algorithm on each setting (e.g., FedAdam has 4 hyper-parameters to tune). Unlike the experiments in the original paper of [Reddi et al., 2021] that are based on million-sized models (e.g., ResNet 18), our experiments are based on billion-sized models (e.g., Llama2-7B), which are much more costly and makes such grid search unaffordable for academic institutions.
> >
> > **Second**, following your suggestions, we have now conducted a more hyper-parameter search based on FedAdam. For example, on the Fed-WildChat dataset, **FedAdam with new hyper-parameters achieve better performance than previous FedAdam and FedAvgM.**
> >
> > [**Table R3.** Results of FedAdam on Fed-WildChat.]
> > |            Method             | MT-1 | Vicuna |
> > |:-----------------------------:|:----:|:------:|
> > |            FedAvgM            | 4.52 |  8.07  |
> > | FedAdam (previous parameters) | 4.54 |  8.03  |
> > |   FedAdam (new parameters)    | 4.68 |  8.35  |
> >
> > **Third**, the seemingly uncorrelated results could result from the fact that there is a gap between the training dataset and testing dataset, making the correlation between training and testing performance less explicit. Unlike in conventional tasks in federated learning that the training and testing datasets are from the same source (e.g., training dataset and testing dataset of CIFAR-10), in the field of LLM, LLMs that are trained on various training datasets are usually tested on the same benchmark (e.g., MT-Bench). In this case, even if a training method can achieve the best training performance by fitting the training data well, it does not necessarily achieve the best testing performance at the same time.

---

> > ### Author Rebuttal · Authors · 2024-08-16
> >
> > **OFI1:** How their datasets fit into the landscape of realistically heterogeneous datasets for FL + LLM research.
> >
> > **Response:** Our datasets are all naturally split by real-world user ID, which reflects realistic heterogeneity in practice. As an example, a user who is a PhD student majoring deep learing would hold significantly different data compared to a user who is a writer. As another example, a user from Japan may uses totally different language compared to a user from France. Therefore, our datasets are realistically heterogeneous datasets for FL + LLM research. Please also refer to our explainations in our responses to **W1**.
> >
> > ---
> >
> > **OFI2:** Why their datasets that partition existing centralized datasets are different from other works that do this. I think the key here is dataset-agnostic partitioning, versus dataset-specific partitioning (which is done in other work, but is not mentioned here).
> >
> > **Response:** Please refer to our responses to **W1**. In general, our datasets are naturally partitioned by real-world user ID, which exhibits natural heterogeneity existing in real-world scenarios. Note that such statements are common in existing benchmark paper in FL [8,9,10]. We will include discussions about both dataset-agnostic partitioning and data-specific partitioning in the revision.
> >
> > ---
> >
> > **OFI3:** Why do you cap the datasets to only users with a certain number of examples? Can the benchmark be altered to instead allow users to experiment with changing the cap themselves, so that if a user wants, they can actually use the entire dataset?
> >
> > **Response:** Please refer to our responses in **W3**. We **have released** the unfiltered version of our datasets. For the second question, the answer is yes.
> >
> > ---
> >
> > **Response to 'Relation To Prior Work'**
> >
> > First of all, we would like to thank the reviewer for the detailed comments and recommending several related works. In the  the following, we would like to address your concerns in detail. And we promise to include these related works in our revision.
> >
> > 1. Our datasets are different from [1] in two aspects: tasks and split mechanism.
> >     - **Different tasks.** Our datasets focus on federated instruction tuning and federated preference alignment, which make unique contributions to the community of federated learning. In contrast, [1] releases four datasets: FedC4, FedWiki, FedBookCO, and FedCCNews, which are for the task of federated pre-training (please refer to page 6 in https://proceedings.neurips.cc/paper_files/paper/2023/file/662bb9c4dcc96aeaac8e7cd3fc6a0add-Paper-Datasets_and_Benchmarks.pdf). Each pre-training data sample is a sequence of text while each data sample in our datasets is a pair of instruction with corresponding response.
> >     - **(2) Different split mechanisms.** Our datasets are split by real-world user ID as in [8,9,10], which exhibit real-world heterogeneity since cross-user is a common setup in federated learning applications. While datasets in [1] are split by tags such as web domain, article, or book, which does not have the cross-user property.
> > 2. Our datasets are different from those in [2,3] from the perspective of **partition machanism.** Dataset in [2] is partitioned according to the tag of 'task type' (76 in total). For example, each client’s instances belong to a single task type; while our datasets are split by real-world user ID. Similarily, dataset in [3] is partitioned according to the tag of programming langauge. However, it is impractical to assume that each client holds only one single task type (e.g., Client A only holds data of translation task while Client B only holds data of summarization task). But rather, practical scenario would be like a user who is a PhD student majoring in machine learning could hold data about coding and academic writing. Even if two users are of the same identity, they could still have evidently different data (e.g., User A prefers concise response while User B prefers informative response). Such real-world properties are hard to be realistically captured by mechanisms in [2,3].
> > 3. Our datasets are different from that in [4] in two aspects: setup and task.
> >     - **Different setup.** Our setup is that clients collaboratively train an LLM; while the setup in [4] is that clients collaboratively train a small model with the help of an LLM on the server side. These two setups are significantly different.
> >     - **Different task.** We focus on instruction tuning and preference alignment; while [4] focuses on next word prediction, which is essentially a pre-training task.
> > 4. Our datasets are different from that in [5] in three aspects: specific task, task coverage, and evaluation.
> >     - **Specific task.** Our task is to train a helpful chatbot that follows humans' instructions, which is a common and popular task in current LLM research; while the tasks in [5] are either mimicing human-human causal talk or limited to text summarization. Therefore, our task is more well-aligned with the current trend of LLM research.
> >     - **Task coverage.** Our datasets also make a unique contribution to releasing the dataset for federated preference alignment, which is not considered by [5].
> >     - **Evaluation.** Our evaluations cover diverse perspectives that follow the trend in the current research of LLM, such as open-ended evaluations (e.g., MT-Bench and Vicuna Bench) and popular benchmarks (e.g., HumanEval, MMLU). However, the evaluations in [5] are still based on metrics in traditional NLP community (e.g., RougeL and Perplexity).
> >
> > Overall, we believe that we make unique contributions to the federated learning community. And we will include this part to improve our paper.
> >
> > ---
> >
> > **Response to 'Documentation'**
> >
> > We have now uploaded the licence to our repository. Regarding future plans, we will continue work on this repository to include more datasets and more federated learning algorithms.
> >
> > ---
> >
> > Thanks again for valuable time. Hope that our responses can well address your concerns.

---

> ### Author Response · Authors · 2024-08-30
>
> Dear Reviewer:
>
> Thanks again for your valuable comments. This is a kind reminder that the discussion period is coming to the end. We have now provided more clarifications, explanations, and experiments to address your concerns. Specifically, we:
> - have clarified our unique contributions by providing more explainations and comparisons.
> - released the unfiltered version of our datasets and conducted corresponding experiments.
> - further tuned the hyper-parameters of FL algorithms.
>
> Please kindly let us know if anything is unclear. We truly appreciate this opportunity to improve our work and shall be most grateful for any feedback you could give to us.

---

### Official Review · Reviewer_Mo2v · 2024-07-21
**Good benchmark for the federated learning community**

**Rating:** 8
**Confidence:** 4
**Correctness:** This paper is technically sound.
**Clarity:** Yes. The paper is well-organized and …

**Review:**

Pros:
- This paper is well-written and easy to follow.
- This paper makes a meaningful contribution to the community of federated learning. It fills the gap that there is currently no realistic benchmark for federated large language models.
- The proposed benchmark covers multiple datasets and tasks. The authors comprehensively demonstrate the properties of these datasets, which benefits future research.
- The authors conduct extensive experiments with many baseline methods to show empirical results for the researchers in the community.

Cons:
- More descriptions on how the fine-tuning methods are implemented with federated learning methods
- The authors should describe more on the considered baseline methods. For example, it should be explained why they choose these baselines and what are the set hyper-parameters.

**Strengths:**

See Pros in Review.

**Additional Feedback:**

N/A

**Documentation:**

The authors provide detailed documentation and Github.

**Limitations:**

This work has no potential negative societal impact.

**Opportunities For Improvement:**

Please continue to update and maintain the GitHub repository as new FL methods and tasks become available. Additionally, it’s highly recommended to consider enabling permissions to allow other users to contribute to the repository, fostering a collaborative and dynamic development environment.

**Relation To Prior Work:**

Yes. This work covers appropriate related works.

**Summary And Contributions:**

This paper proposes the first realistic benchmark for the community of federated large language models. The benchmark includes four datasets for instruction tuning and preference alignment. The authors conduct extensive experiments on this benchmark with 8 classical baselines. They also show explorations on multilingual collaborations and differential privacy.

---

> ### Author Rebuttal · Authors · 2024-08-16
>
> **W1:** More descriptions on how the fine-tuning methods are implemented with federated learning methods
>
> **Response:** Thanks for the advice. In our codebase, the implementation of fine-tuning methods and federated learning methods are decoupled so that they can be combined without much effort. In this case, researchers from the FL community could implement their FL methods to couduct FedLLM experiments as simply as what they do for conventional image classification tasks.
>
> For LLM fine-tuning methods, we treat them as basic Trainer classes in python code, which will be launched by each client for local model training. For example, instruction tuning (i.e., supervised fine-tuning) is implemented in a SFTTrainer class, which auto-regressively computes the loss on the response of each example (instruction paired with response). Likewise, direct preference optimization is implemented in a DPOTrainer class, which computes the DPO loss to better align with the preferred response of each example.
>
> For federated learning methods, we treat them differently according to their operation side (client or server). For client-side methods, taking FedProx as an example, it inherits the LLM fine-tuning basic class (which could be SFTTrainer or DPOTrainer based on the task) first and then adds an additional loss that regularizes the distance between local and global models. For server-side methods, taking FedAvgM as an example, it simply modifies the process of model aggregation on the server side, while using the basic trainer class on the client side.
>
> Thanks for your advice. We will include these descriptions in the revision.
>
> ---
>
> **W2:** The authors should describe more on the considered baseline methods. For example, it should be explained why they choose these baselines and what are the set hyper-parameters.
>
> **Response:** Thanks for the advice. The chosen baselines are all influential works in the community of federated learning (e.g., with high citations) and they cover two types (classified by operation on client or server side) of federated learning methods.
>
> - FedAvg [1] is the basic federated learning method.
> - Client-side methods: FedProx [2] applies an l2 regularization term between the local model and global model during the training of local model on the client side. SCAFFOLD [3] introduces a control variate that corrects the gradient of local model on the client side.
> - Server-side methods: FedAvgM [4] introduces simple momentum for updating the global model on the server side. FedAdagrad, FedYogi, and FedAdam [5] introduce adaptive optimization methods for updating the global model on the server side.
>
> The hyper-parameters are set as follows:
>
> [**Table R2.** Hyper-parameters of baselines.]
> |  Baseline   |                     Hyper-parameters                      |
> |:----------:|:---------------------------------------------------------:|
> |  FedProx   |                        $\mu=0.01$                         |
> |  FedAvgM   |                       momentum=0.5                        |
> | FedAdagrad |   $\beta_1=0$, $\beta_2=0$, $\eta_g=1e-3$, $\tau=1e-3$    |
> |  FedYogi   | $\beta_1=0.9$, $\beta_2=0.99$, $\eta_g=1e-3$, $\tau=1e-3$ |
> |  FedAdam   | $\beta_1=0.9$, $\beta_2=0.99$, $\eta_g=1e-3$, $\tau=1e-3$ |
>
> [1] McMahan, Brendan, et al. "Communication-efficient learning of deep networks from decentralized data." Artificial intelligence and statistics. PMLR, 2017.
>
> [2] Li, Tian, et al. "Federated optimization in heterogeneous networks." Proceedings of Machine learning and systems 2 (2020): 429-450.
>
> [3] Karimireddy, Sai Praneeth, et al. "Scaffold: Stochastic controlled averaging for federated learning." International conference on machine learning. PMLR, 2020.
>
> [4] Hsu, Tzu-Ming Harry, Hang Qi, and Matthew Brown. "Measuring the effects of non-identical data distribution for federated visual classification." arXiv preprint arXiv:1909.06335 (2019).
>
> [5] Reddi, Sashank J., et al. "Adaptive Federated Optimization." International Conference on Learning Representations.
>
> ---
>
> **Suggestion:** Please continue to update and maintain the GitHub repository as new FL methods and tasks become available. Additionally, it’s highly recommended to consider enabling permissions to allow other users to contribute to the repository, fostering a collaborative and dynamic development environment.
>
> **Response:**
> Thanks for your advice. We will continue to update and maintain the repository and welcome other users to contribute.
>
> ---
>
> Overall, we hope that our responses can fully address your comments and will be grateful for any feedback.

---

### Official Review · Reviewer_ziZv · 2024-07-23
**Review of FedLLM**

**Rating:** 7
**Confidence:** 5
**Correctness:** Yes.
**Clarity:** Yes

**Review:**

This work introduces a realistic benchmark for federated learning of LLM. This benchmark includes multiple 8 training methods, training datasets, and evaluation metrics. Furthermore, it covers diversities as well: language, quality, quantity, length, and preference. Specific pros and cons, please check Strengths and Opportunities For Improvement.

**Strengths:**

- This paper is well-structured and well-written.

- The authors make detailed descriptions of their datasets in FedLLM-Bench and provide a corresponding in-depth examination to show the real-world properties of them.

- This paper introduces a novel benchmark as there is no existing realistic benchmark for federated learning of large language models.

- A comprehensive empirical study is provided. The authors include experiments on standard setups and also make some novel explorations on multilingual collaboration.

**Additional Feedback:**

N/A

**Documentation:**

Yes.

**Ethics:**

No.

**Limitations:**

No potential negative societal impact of this work.

**Opportunities For Improvement:**

- Though the authors implement several representative baselines, they did not mention how to quickly implement new algorithms based on their framework.

- Perplexity is a common metric for evaluating a data sample in language modeling. It would be better to show this property of the considered datasets.

**Relation To Prior Work:**

Yes

**Summary And Contributions:**

This paper addresses the issue that existing literature on federated learning of large language models often relies on artifitially partitioning existing centralized datasets. It proposes the first realistic benchmark with four realistic decentralized datasets and two common tasks for FedLLM. Various visualizations are demonstrated to show the realistic properties of the datasets. Experimental results on each dataset with eight baselines and some additional study are provided.

---

> ### Author Rebuttal · Authors · 2024-08-16
>
> Thank you for your time reviewing our paper and your recognition. Here are our responses to your comments in detail.
>
> ---
>
> **W1:** Though the authors implement several representative baselines, they did not mention how to quickly implement new algorithms based on their framework.
>
> **Response:** Thanks for this valuable advice. We would like to provide a corresponding introduction here and will include this in the revision. We can roughly divide most FL algorithms into two steps: (1) clients receive information from the server and start local model training; (2) server collects information from the server and updates the global model.
> 1. For algorithms that focus on the first step (e.g., FedProx), users only need to create a new Trainer class that inherits the basic Trainer. For example, to achieve FedProx on federated instruction tuning, the only thing to do is to inherit the SFTTrainer and overwrite the 'compute_loss()' function by adding a regularization loss, which takes less than 10 lines of code. Please refer to the 'federated_learning/fed_local_sft.py' file for details.
> 2. For algorithms that focus on the second step (e.g., FedAvgM), users only need to modify the aggregation function in the 'federated_learning/fed_global.py' file. For example, to achieve FedAvgM, the only thing to do is to add content about computing the momentum and aggregating with the momentum, which takes less than 5 lines of code.
>
> ---
>
> **W2:** Perplexity is a common metric for evaluating a data sample in language modeling. It would be better to show this property of the considered datasets.
>
> **Response:** Thanks for the advice. We have now computed the perplexity for all datasets and report the median value in the following table. We will include this in our revision.
>
> [**Table R2.** Results of FedAdam on Fed-WildChat.]
> |  Dataset   | Fed-Aya | Fed-ChatbotIT | Fed-WildChat | Fed-ChatbotPA |
> |:----------:|:-------:|:-------------:|:------------:|:-------------:|
> | Perplexity |  3.941  |     2.881     |    3.395     |     2.868     |
>
> ---
>
> Overall, we hope that our responses can fully address your comments and will be grateful for any feedback.

---

### Official Review · Reviewer_GdVS · 2024-07-24

**Rating:** 8
**Confidence:** 4
**Correctness:** Yes. The claims made in the submissio…
**Clarity:** Yes. The paper is well written.

**Review:**

This work is clearly presented and is of high quality. It makes sufficient contributions for the community of federated learning. See the pros and cons in the part of strengths and weaknesses.

**Strengths:**

- The targeted topic is interesting and up-to-date. The paper is easy to follow with a good structure.
- The proposed safety attack method is effective, which can break the safety alignment of FedIT without being defended by many existing defense methods.
- The proposed defense method is quite new in FL, which can automatedly generate data to achieve defense.
- Experiments on diverse training datasets and benchmarks show the effectiveness of the proposed attack and defense methods.

**Additional Feedback:**

NA

**Documentation:**

Yes.

**Ethics:**

Yes.

**Limitations:**

See the Opportunities For Improvement.

**Opportunities For Improvement:**

- The authors should explain the reasons for filtering the datasets and I suggest that the unfiltered version should be also released
- Most of the results in this paper are based on Llama2-7B. What will the observations be if we are using different LLMs?

**Relation To Prior Work:**

Yes.

**Summary And Contributions:**

This paper proposes FedLLM-Bench, the first realistic benchmark for federated learning of large language models especially for the tasks of federated instruction tuning and federated preference alignment. It proposes four realistic federated datasets which are split by real-world user IDs. The proposed datasets exhibit diversity across multiple aspects including languages, quantities, and qualities. Sufficient experiments are conducted to show the performance of several baselines on the proposed benchmark.

---

> ### Author Rebuttal · Authors · 2024-08-16
>
> Thank you for your time reviewing our paper and your recognition. Here are our responses to your comments in detail.
>
> ---
>
> **W1:** The authors should explain the reasons for filtering the datasets and I suggest that the unfiltered version should be also released.
>
> **Response:**
> Thanks for the advice.
>
> First of all, in the following, we would like to explain why we choose the current scale, and we hope to gain your understanding. The reasons are two-folded.
>
> - Such operation is inspired by experience from centralized training of LLMs that iterating too many times on training samples could affect the performance. In centralized training of LLMs [1], the number of epochs is often set as a small number (e.g., 3) and the batch size is often set as a moderate number (e.g., 16, 32, 64, 128). Training too much on data samples or using a small batch size could lead to bad performance. Therefore, in our federated learning setting, we need to consider this similar issue. Let's assume that we set the number of local training steps as 10 and a batch size 16. Then, for a client with few training samples (e.g., only 1). At each round, the data of this client will be iterated for 10 epochs, which is improper according to experience from centralized training. Therefore, we choose to filter out clients with too few samples.
>
> - Choosing such a moderate scale makes the training affordable for a wide range of researchers, which promotes the development of this field by avoiding it becoming the game of only a few players (e.g., large companies). Without filtering, the number of clients could be over 10,000, indicating that we need to run 10,000+ large language models (or their adapters) on their compute resources, which is unaffordable for many researchers.
>
> &nbsp;
>
> Second, following your suggestions, we have now released the unfiltered version of our datasets (see the statistics in Table R1) and conduct initial experiments on top of it. Due to limited time, here, we sample 300 clients of Fed-WildChat for the experiments, where some of the clients hold only a single sample. From Table R2, we still see that federated learning methods consistently and evidently outperform local training, indicating the effectiveness of joining collaboration.
>
> [Table R1. Data distribution of unfiltered datasets]
> | Dataset Name | Fed-Aya | Fed-ChatbotIT | Fed-WildChat | Fed-ChatbotPA |
> |:--------:|:-----:|:---------:|:--------:|:--------:|
> |# Clients (Total)|1456|10996|181063|10996|
> |# Samples (Total)|202364|23294|899215|23294|
> |# Samples (Client)|139±605|2±6|5±57|2±6|
>
> &nbsp;
>
> [**Table R2.** Results on unfiltered Fed-WildChat.]
> | Method | MT-1 | Vicuna | Ref-GPT4 |
> |:--------:|:-----:|:---------:|:--------:|
> |Local|4.15|7.03|4.50|
> |FedAvg|4.61|8.03|5.81|
> |FedProx|4.69|7.98|5.98|
> |Scaffold|4.48|7.95|5.83|
> |FedAvgM|4.63|8.24|5.99|
> |FedYogi|4.68|7.97|5.47|
> |FedAdagrad|4.46|7.98|5.55|
> |FedAdam|4.68|8.11|6.16|
>
> &nbsp;
>
> Overall, we believe that the filtered version would contribute to involving more researchers in this field. Meanwhile, we agree that the unfiltered version could also play its role, which has been released following your suggestions.
>
> [1] https://github.com/lm-sys/FastChat/tree/main
>
> ---
>
> **W2:** Most of the results in this paper are based on Llama2-7B. What will the observations be if we are using different LLMs?
>
> **Response:** Thanks for your advice. Following your suggestions, we have now conducted experiments based on Mistral-7B. The results are presented in the following. From the table, we can still see clear gap between federated learning methods and local training, indicating the evident benefits brought by collaboration.
>
> [**Table R3.** Results based on Mistral-7B.]
> |   Method   | MT-1 | Vicuna | Ref-GPT4 |
> |:----------:|:----:|:------:|:--------:|
> |   Local    | 5.23 |  8.33  |   5.47   |
> |   FedAvg   | 6.08 |  8.91  |   7.21   |
> |  FedProx   | 6.18 |  8.88  |   6.91   |
> |  Scaffold  | 6.24 |  8.85  |   6.85   |
> |  FedAvgM   | 6.20 |  8.99  |   7.14   |
> |  FedYogi   | 6.31 |  9.06  |   7.00   |
> | FedAdagrad | 6.28 |  9.11  |   6.92   |
> |  FedAdam   | 6.41 |  8.88  |   7.35   |
>
> ---
>
> Overall, we hope that our responses can fully address your comments.

---

### Decision · Program_Chairs · 2024-09-26

**Decision:**

Accept (Poster)

**Comment:**

In this submission, the authors present a new benchmark for federated learning of LLMs. The reviewers acknowledge that this submission has several strong aspects, and benchmarking FL of LLMs is very important.

However, after carefully reading all the review comments, the rebuttals from authors and checking the submission, I suggest that the authors should improve this work further in the camera ready version.

The important reason for such decision is about the most important contribution of this submission.
As featured by author themselves, the most novel part of the proposed benchmark is "realistic", not artificially. However, as pointed by reviewer and discussed among review and authors, the datasets are split by user ID. The authors argue that "such definition follows the statements in several previous benchmark papers" and they are also accepted by NeurIPS, so such method is realistic. This is not the case. For example, let's say there are three organizations, they hire several annotators to label their data and give annotation guideline. In this case, the bias introduced by organization's guideline might be larger than the bias introduced by annotators, not mention that annotators hired by different organizations can be the same (i.e., different user ID can be the same annotator). Such claim about realistic is not convincing enough.

This is a promising submission. Hope the comments and discussions are helpful, and authors can make this submission convincing and strong.